# De novo variants in the splicing factor gene *SF3B1* are associated with neurodevelopmental disorders

SF3B1 is an essential and ubiquitous splicing factor that plays a pivotal role in the early steps of pre-mRNA splicing. Recurrent somatic missense mutations in *SF3B1* are frequent in cancers, but no constitutional variant has been reported so far. We describe here a cohort of 26 individuals with neurodevelopmental disorders, harbouring *SF3B1* constitutional heterozygous variants that appeared mostly de novo. Patients present with a global developmental delay, associated with variable neurological and facial dysmorphic traits. A dichotomy may emerge between patients harbouring predicted loss of function ($n = 9$) and missense variants ($n = 17$), the latter being associated with a more severe and syndromic phenotype, including heart and gastrointestinal anomalies. We focused on de novo *SF3B1* missense variants, which were largely distinct from those reported in cancer. Functional complementation assays show that de novo *SF3B1* missense variants did not cause a loss of function of the protein. Targeted and genome-wide analysis of RNA splicing reveal that they affect canonical and alternative splicing more moderately than somatic variants, and subtly modify the splicing of many transcripts. These findings place SF3B1 among the rare U2 snRNP components implicated in both cancer and neurodevelopmental disorders, highlighting its critical and multifaceted role in human disease.

RNA splicing is a crucial process in eukaryotes where introns are excised from a nascent pre-messenger RNA (pre-mRNA), and adjacent exons are joined to form a mature mRNA. Alternative RNA splicing is a complex and highly regulated mechanism that widely diversifies the proteome by creating multiple proteins from a single gene[1]. In humans, most introns are recognized and excised by the major spliceosome, which consists of a large and dynamic ribonucleoprotein (RNP) complex including small nuclear RNAs (U1, U2, U4/U6 and U5), in interaction with auxiliary proteins. The spliceosome assembly relies on sequential and dynamic RNA-RNA interactions between U2 and U6 small nuclear RNA (snRNA), and between snRNAs and intronic splice sites[2]. Disruption in spliceosome components due to genetic variants lead to disorders referred to as spliceosomopathies[3].

SF3B1 (Splicing factor 3b, subunit 1) is an essential, ubiquitously abundant protein, which plays a pivotal role in the early steps of pre-mRNA splicing, by stabilising the U2 snRNP complex at branch point (BP) sequences. SF3B1 is the largest subunit of SF3b, a heptameric protein complex of U2 snRNP. SF3B1 comprises an unstructured N-terminal domain and a conserved highly structured C-terminal domain called HEAT (Huntington Elongation Factor 3 protein, phosphatase 2 A and the yeast PI3-kinase TOR1), composed of 20 repeats[4]. *SF3B1* is the most frequently mutated RNA splicing factor across cancers, notably in haematological malignancies[5] including myelodysplastic neoplasms[6] and chronic lymphocytic leukaemia, as well as in solid tumours, such as uveal melanoma[7], breast and pancreatic cancers. Until now, only somatic cancer-associated variants of *SF3B1* have been reported. They are essentially missense heterozygous variants

✉ e-mail: delphine.bernard@univ-brest.fr

resulting in recurrent amino acid substitutions in the H4-H8 repeats of the HEAT domain, with K700E hotspot being the most frequent[4]. RNAseq studies conducted in various cancers have shown that *SF3B1* somatic variants are change-of-function variants that alter canonical and alternative splicing through the recognition of cryptic BP and the use of cryptic or alternative 3' splice sites[7–9], leading to a remodelling of approximately 1% of the transcriptome. As a result, novel transcripts involved in a diversity of cellular pathways are generated, which may be recognized by nonsense mediated mRNA decay (NMD) or may lead to novel protein isoforms[8,10]. Although all somatic *SF3B1* variants alter the proper recognition of BP sequences, the missplicing pattern can partly vary with the variant nature, and some variants of *SF3B1* have been associated with distinct clinical outcomes, highlighting differential functional impact of cancer-associated *SF3B1* variants[11,12].

Defects in spliceosome genes, including components of U2 snRNP, have been described in various syndromes, notably neurodevelopmental disorders (NDD)[13,14]. NDD of genetic origin are a heterogeneous group of conditions that manifest during childhood, including intellectual disability (ID) and autism spectrum disorder (ASD), collectively affecting 3% of the population[15–18]. Exome or genome sequencing has become a standard procedure for the aetiological diagnosis of these disorders[19]. More than 1500 genes are known to be involved in NDD[15], some of them encode components or regulators of the spliceosome[20,21]. The latter includes genes encoding core or auxiliary components of the U2 snRNP complex to which SF3B1 belongs, such as *U2AF2*[20], the U2 snRNA *RNU2-2*[22] or *PUF60*[23]. Deleterious variants in *PUF60* are responsible for Verheij syndrome (MIM: #615583) in which ID is accompanied by microcephaly and various skeletal malformations[13,20,23,24].

Here, we report a cohort of 26 individuals with NDD, harbouring *SF3B1* variants, which appeared mostly de novo and are largely distinct from those commonly found in cancer. Given the prevalence of de novo variants in developmental disorders[25] and the essential function of *SF3B1* in pre-mRNA splicing, we postulated that pathogenic de novo variants affecting the function of SF3B1 could be implicated in NDD. To decipher the molecular and cellular consequences of these de novo *SF3B1* variants, we conducted targeted analysis and RNA sequencing (RNAseq) in dedicated cellular models expressing the *SF3B1* variants of interest. Our findings provide compelling evidence supporting an association between de novo *SF3B1* variants and NDD.

## Results

### Patients carrying predicted loss-of-function and missense variants of *SF3B1* present with a neurodevelopmental syndrome

The 26 individuals participating in the study were included based on the identification of a heterozygous *SF3B1* variant identified by exome or genome sequencing, suspected to be pathogenic according to in silico predictions, frequency in population databases and/or inheritance and segregation in the family. In total, we identified 25 distinct *SF3B1* variants that mostly occurred de novo, in 26 unrelated individuals, with one of the variants being recurrent. Our cohort comprised 10 female and 16 male patients with an age range of 1–24 years at the time of inclusion in the cohort. Clinical data are presented in Fig. 1, Table 1 and Supplementary Data 1. We did not find any other genetic variant that may explain the phenotype of the patients (Supplementary Data 1), except for patient 1, who carries a variant of uncertain significance in the *LARP1* gene, for patient 19, who is heterozygous for an inherited variant in *CACNA1A*, and for patient 22, who carries an inherited variant of uncertain significance in *RORA*.

The series encompasses 16 missense variants and 9 putative loss-of-function (pLoF) variants, including three splicing variants, three nonsense, one frameshift insertion and two frameshift deletions (Fig. 2A). All variants were absent from the general population databases (gnomAD V4.1.0 - NonUKB), except for p.R397H found in one individual, arguing against simple polymorphism. All variants were

private, except for the p.R939C missense variant, found in two unrelated individuals. The use of several splicing prediction tools (MaxEntScan[26], SpiP[27], dbscSNV[28], SpliceAI[29]) does not provide strong evidence that missense variants may disrupt an ESE or favour usage of a novel splice site (Supplementary Table 1). We investigated the 16 missense variants using in silico pathogenicity predictors (SIFT, PolyPhen2, MutationTaster, CADD, REVEL, Alphamissense) and found that most were predicted to be pathogenic (Supplementary Table 2). These variants often affect highly conserved amino acids in constrained SF3B1 regions, as indicated by the missense tolerance ratio (MTR) and CADD scores (Fig. 2C). Based on ACMG (American College of Medical Genetics and Genomics) guidelines and presuming a gene-disease association, most variants were classified as Likely Pathogenic or Pathogenic, except four of them (classified as of uncertain significance), namely p.N829S, p.A1229G, p.R736C and p.R397H (Supplementary Table 3).

Almost all (23/26) affected individuals exhibited at least one neurodevelopmental abnormality, the most frequent being language delay (21/26). Motor delay was found in 18/24 individuals. Intellectual disability (ID) was present in 9/15 individuals, some cases being too young (8/26) to assess. The severity of ID was mainly mild to moderate, only two individuals presenting with severe ID (12 and 21). Individuals 9 and 10, although not presenting with neurodevelopmental delay, suffered from spastic quadriplegia. Seizures were reported in 13 individuals, with variable ages of onset and types, but infantile spasms were recurrent (3/13). Other notable neurological features included hypotonia, which was present in eleven patients. When performed, brain MRI did not identify recurrent abnormalities, except a corpus callosum anomaly in two individuals (3 and 22), and posterior fossa anomaly in two other cases (2 and 8).

Dysmorphic features were found in 20/26 individuals, although we could not identify a typical facial gestalt (Fig. 1B). Some non-specific signs, such as low-set ears, were recurrent, as well as cupid's bow lips, found in two patients. The most notable feature was palatal abnormality, as 12 patients exhibited cleft palate or a high-arched palate. Ophthalmologic findings were not specific.

Heart anomalies were found in 8/25 individuals, mostly atrial and ventricular septal defects (patients 15, 16, 17, 20). The individual presenting the most severe cardiac condition was case 6, who was diagnosed with tetralogy of Fallot.

Intrauterine growth retardation was present in five individuals, and six individuals exhibited postnatal short stature. Head circumference, when available, was normal in 15 individuals, whereas seven presented with microcephaly. The most recurrent and notable findings were feeding challenges. Eight patients suffered from feeding difficulties, with the need for a gastrointestinal tube during infancy for six of them, resulting in growth delay or low weight.

Almost all variants arose de novo, except in three cases (P21, P22, P23) that were all pLoF variants. In two of these cases, the parent carrying the variant was symptomatic. In individual 21, the variant p.E809Dfs*7 was inherited from the mother, who had learning difficulties. His sister, also carrying the variant, presented with developmental delay, autism spectrum disorder and learning difficulties. One maternal uncle had adaptive schooling and was followed in psychiatry. Two maternal aunts stopped school early, and one first cousin, deceased, presented with seizures and developmental delay. In individual 22, the father carrying the variant p.V695Wfs*34 presents with developmental delay and discontinued formal education at the age of 14. In only one case (individual 23), the variant p.R166* was inherited from an asymptomatic father; however, segregation analysis revealed that the variant had arisen de novo in the father. Finally, in individual 24, the segregation was incomplete due to the absence of paternal DNA.

To investigate a putative genotype-phenotype correlation, we first performed an unsupervised clustering of clinical features. This

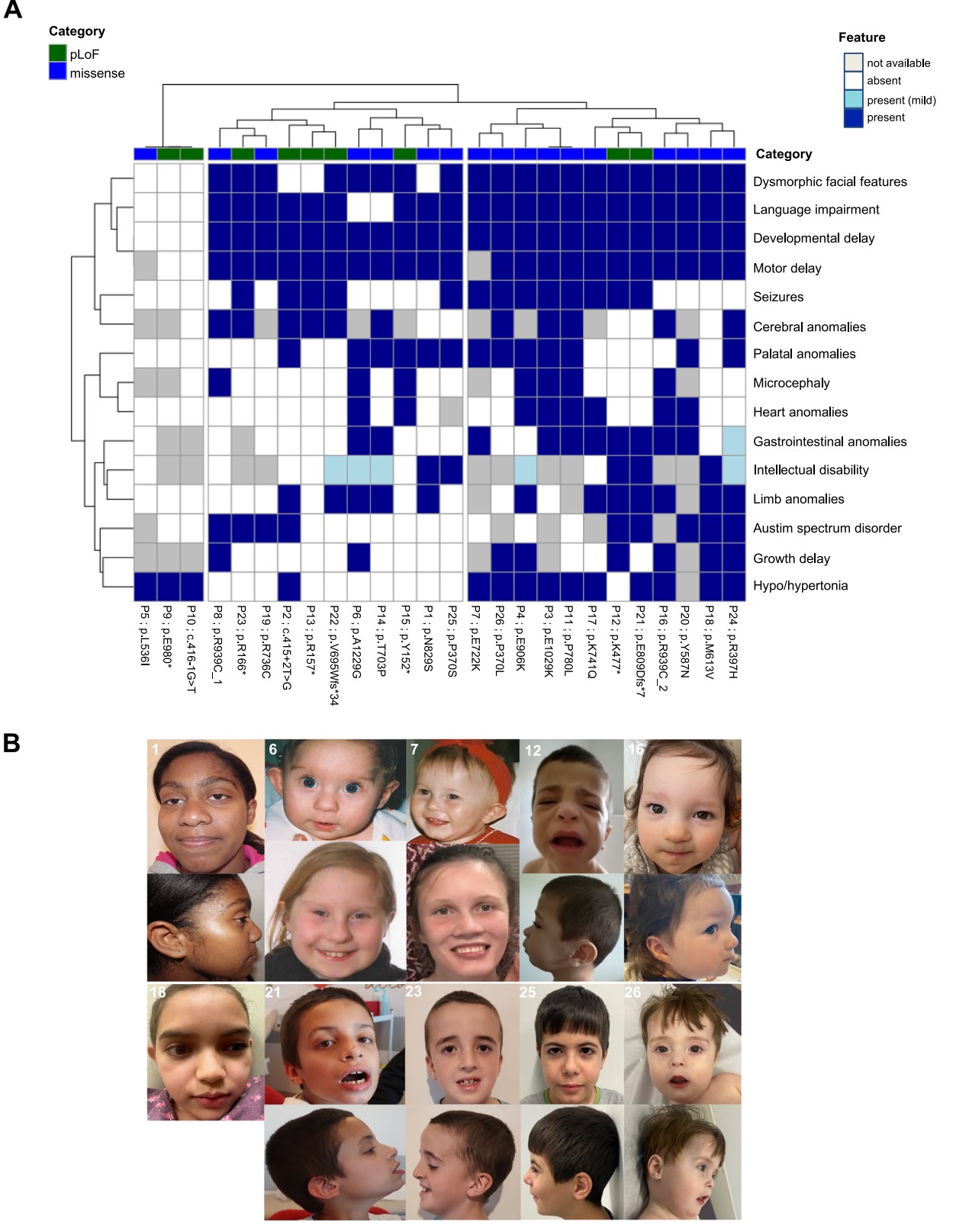

**Fig. 1 | Individuals carrying predicted loss-of-function (pLoF) and missense variants of *SF3B1* present with a neurodevelopmental syndrome. A** Hierarchical clustering of the clinical features (rows) of the cohort's patients (columns). Categorical data was converted to 0–1 scale: 0 = feature absent (white box), 0.5 = feature present with mild severity (light blue box), 1 = feature present, moderate to severe (blue box). Missing values are shown in grey. Columns are coloured based on the variant category (green: putative loss-of-function, blue: missense). Individuals carrying missense *SF3B1* variants may be associated with a more severe phenotype in comparison to *SF3B1* pLoF variants. **B** Photographs of 10 individuals, referred to as P1, P6, P7, P12, P16, P18, P21, P23, P25 and P26 in (**A**).

**Table 1 | Summary of the main clinical signs of individuals carrying SF3B1 variants**

| | | Total missense | Total pLoF | Total |
|---|---|---|---|---|
| Growth | Short stature | 7/13 | 1/7 | 8/20 |
| Neurodevelopment | Microcephaly | 6/14 | 1/8 | 7/22 |
| | Developmental delay | 16/17 | 7/9 | 23/26 |
| | Language delay | 14/17 | 7/9 | 21/26 |
| | Motor delay | 13/15 | 5/9 | 18/24 |
| | Intellectual disability | 6/9 | 3/6 | 9/15 |
| | Seizures | 7/17 | 6/9 | 13/25 |
| | Hypotonia | 9/15 | 2/9 | 11/24 |
| Malformations | Dysmorphic features | 15/17 | 5/9 | 20/26 |
| | Cleft palate / palatal abnormality | 11/17 | 1/9 | 12/26 |
| | Hand/foot abnormalities | 7/14 | 4/9 | 11/23 |
| | Gastrointestinal abnormalities | 9/14 | 2/9 | 11/23 |
| | Cardiac abnormalities | 7/16 | 1/9 | 8/25 |

clustering suggested the presence of three patient groups, which appear to differ in terms of phenotype severity and variant type (Fig. 1A). The left cluster comprised three patients presenting only with abnormal muscle tone, two of whom carried a pLoF variant. In the right cluster, where most patients harboured a missense variant, we observed a trend toward more severe phenotypes and a higher frequency of congenital anomalies, suggesting a possible correlation between the type of variation (missense or pLoF) and the clinical outcome. To mitigate potential bias arising from missing data, we performed unsupervised clustering after excluding clinical variables with missing information in more than 30% of patients. This analysis yielded comparable results, with missense variants consistently clustering together (Supplementary Fig. 1). More specifically, short stature was mostly found in individuals with missense variants (7/13 vs 1/7), as well as gastrointestinal abnormalities (mainly feeding difficulties, with the need of gastrostomy, 9/14 vs 2/9)(Table 1). Microcephaly was present in 6/14 individuals with missense variants, and in only one individual with a pLoF variant. Congenital anomalies, especially cardiac malformations, were also more frequent in patients with missense variants (7/16 vs 1/9), as well as palatal defects (11/17 vs 1/9). The only variants that were inherited were pLoF, also suggesting that they could be associated with a less severe phenotype or a variable expressivity. Overall, these findings suggest that missense variants could have a functional effect different from pLoF variants.

## SF3B1 variants associated with neurodevelopmental disorders are distinct from cancer-associated SF3B1 variants

The missense variants affect highly conserved amino acid positions, which are distributed along the HEAT domain except for R397 and P370 (Supplementary Fig. 2 and Fig. 2D). Six missense variants impact amino acids that are clustered into H4 to H8 repeats, like the most frequent somatic variants (including p.R625H, p.K666N and p.K700E). To better understand the consequences of the missense variants on SF3B1 function, we mapped the position of the residues on the SF3B1 structure from a human activated spliceosome (mature Bact complex)(PDB:5Z56)(Fig. 2D). SF3B1 is required to stabilise the U2snRNA-pre-mRNA duplex at the BP region, making this step of the splicing cycle relevant to study. Remarkably, the substituted residues are mainly located on the same side of the HEAT domain, facing the pre-mRNA. Only three residues (R736, E722 and R939) face outward. Notably, the residues T703, R736, K741 and P780 are in a pocket

(namely the RNA path) that faces the three most affected residues in cancer, i.e. R625, K666 and K700 (Fig. 2D). We aimed to predict possible consequences of the substitutions on SF3B1 intramolecular and intermolecular interactions using Pymol. For the most part, the substituted residues are predicted to affect interactions within a HEAT repeat helix, between different HEAT repeat helices or with SF3B1 partners (Fig. 2E and Supplementary Fig. 3). Some substitutions, such as T703P, are predicted to increase intramolecular interactions, rigidifying the structure locally, while others may disrupt local interactions (such as N829S, R939C). Glutamic acids at positions 906 and 1029 are close to the pre-mRNA, and the change of charge induced by the substitution by a lysine may affect the positioning of the pre-mRNA. The proline at position 780 is in a close proximity with the pre-mRNA and the protein RBMX2 and its substitution by a leucine may affect these interactions. Remarkably, half the missense variants lead to substitution of residues located close to residues directly involved in the interaction with DDX42 and DDX46[30,31] helicases, which act in the early steps of spliceosome assembly. In particular, K741 has been shown to directly interact with DDX42 and DDX46. Since the spliceosome undergoes intense compositional and structural changes during the splicing cycle, some of the substituted residues may be involved in other important interactions, beyond the Bact complex.

Notably, out of the 15 missense variants, 8 were not reported at all in the Catalogue of Somatic Mutations in cancer database (COSMIC), while 7 of them were reported in only a few cases of tumour samples or cell lines (range of 1 to 4 reported cases)(Supplementary Table 4), including p.T703P, p.E722K, p.R736C and p.K741Q. This suggests that the SF3B1 variants associated with NDD are predominantly distinct from common cancer-associated SF3B1 variants.

## SF3B1 missense variants associated with a neurodevelopmental disorder do not cause a loss-of-function

We aimed to determine the functional consequences of de novo SF3B1 missense variants, the most frequent variants in our series. To investigate both change-of-function (as for cancer-associated variants) and loss-of-function hypotheses, we followed the same targeted approach previously used by our group for other SF3B1 variants[32]. Importantly, somatic SF3B1 variants are neomorphic variants that lead to splicing alterations distinct from those observed upon SF3B1 silencing (siSF3B1/shSF3B1)[33,34]. The principle of the analysis was to express the SF3B1 variants of interest in a same cellular model in which the expression of endogenous SF3B1 was silenced, in order to study the ability of each SF3B1 variant to suppress the splicing and cell proliferation defects caused by endogenous SF3B1 silencing. Considering the high occurrence of SF3B1 somatic mutations in haematological neoplasms, we decided to use erythroleukemia K562 cells as a cellular model to study all de novo SF3B1 missense variants, in comparison to well established controls developed in this cell model. Indeed, several RNAseq and functional studies have been performed in K562 cells harbouring cancer-associated SF3B1 mutations, both by our group[32] and by others[35–37].

Four additional missense variants were identified in patients after experimental work was near completion, for this reason we studied a total of 11 (out of 15) SF3B1 missense variants. The steady-state levels of each variant protein, tagged with a FLAG epitope on N-terminus, were similar to the wild type counterpart, indicating that each variant preserves the stability of the protein (Supplementary Fig. 4). We then investigated specific exon skipping events known to be altered upon SF3B1 silencing, in particular in DUSP11 (exon 6) and RBM5 (exon 16)[32]. As a control, we included a condition in which an inactive isoform of SF3B1 was expressed, namely SF3B1ins, which we previously characterized as an aberrant transcript produced upon expression of K700E[32]. Moreover, should the E980* variant escape the NMD pathway, thereby leading to the translation of a truncated protein, we sought to investigate the functionality of the resulting polypeptide. As

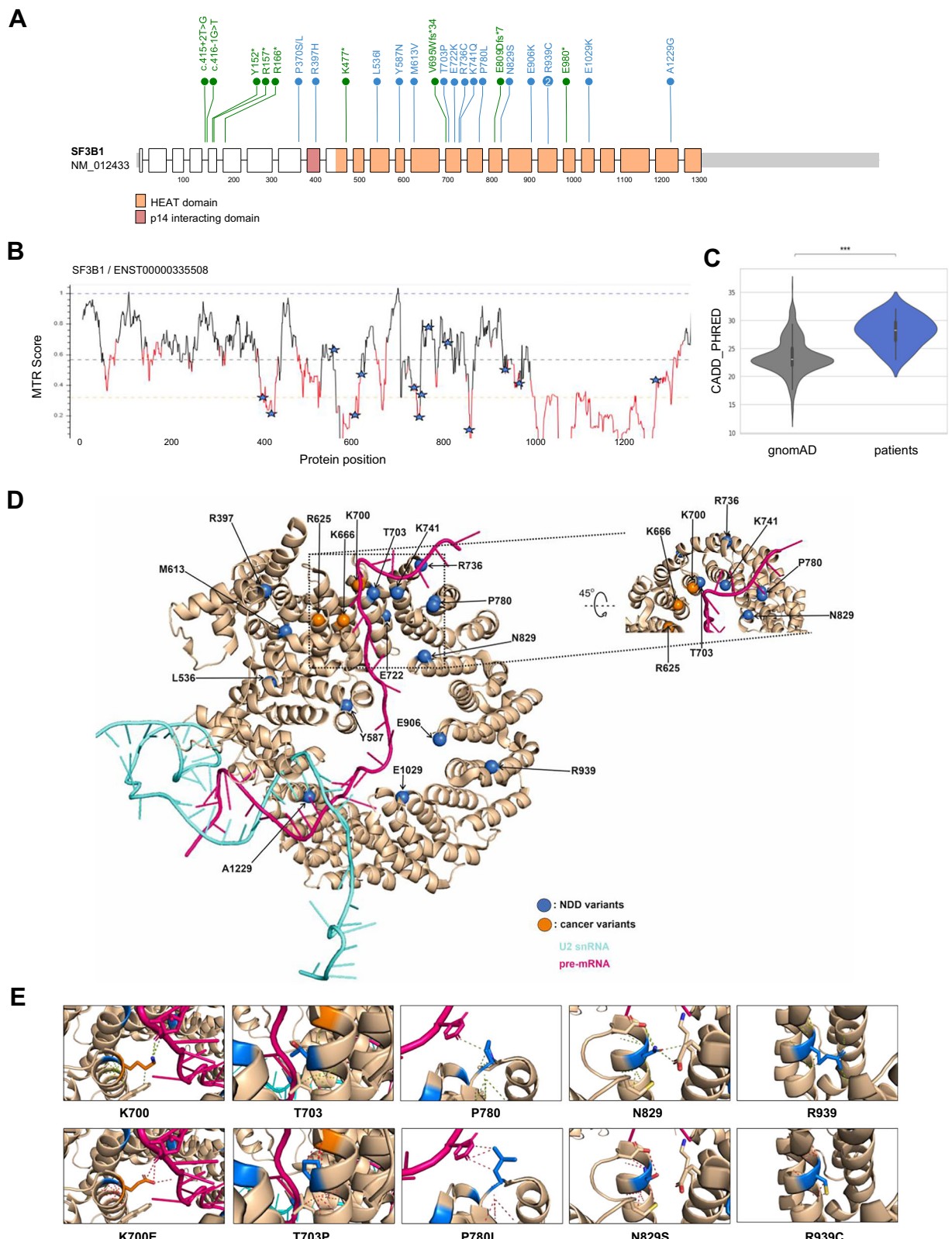

expected, expressing the truncated E980* protein did not suppress the splicing defects caused by *SF3B1* silencing, confirming the essential nature of the last third of the HEAT domain. In contrast, expressing each missense variant restored splicing proficiency, indicating that none of the eleven missense variants led to a loss-of-function phenotype (Fig. 3A, B and Supplementary Fig. 4). To confirm these results, we used the same readouts in an alternative cell model, HEK293T, in which

we expressed eight missense variants (Supplementary Fig. 5) and obtained similar findings.

We then studied the ability of selected *SF3B1* variants to suppress the proliferation defect caused by *SF3B1* silencing. Four *SF3B1* variants (E722K, P780L, N829S and E980*) were expressed under the control of a doxycycline-inducible promoter in K562 cells in which endogenous *SF3B1* was silenced. While expressing the E980* truncated protein was

**Fig. 2 | Position and distribution of the *SF3B1* missense variants reported in the patient cohort. A** Distribution of *SF3B1* variants identified in this study. Missense variants are indicated in blue, loss-of-function variants in green. **B** Missense tolerance ratio (MTR) plot of SF3B1, which quantifies selective pressure against missense variants in the human population. MTR = 1 indicates neutrality. Red segments mark windows with exome-wide FDR < 0.10 for deviation from MTR = 1. Stars represent cohort missense variants. **C** Comparison of CADD scores between patients' (blue, $n$ = 16) and rare gnomAD variants (grey, $n$ = 532; allele frequency <$10^{-4}$, gnomAD V2.2.1). Violin plots show CADD_PHRED distribution. White line = median; black box = interquartile range (IQR) (25th–75th percentiles). Whiskers extend to 1.5 × IQR beyond the quartiles; the violin width reflects the kernel density. $p$ = 5.32 × $10^{-6}$ ( < 0.0001, ****; two-sided Student's *t* test. **D** Localisation of missense affected residues in the SF3B1 HEAT domain (closed conformation), including pre-mRNA

(pink) and U2 snRNA (cyan) interaction network at the branch point (Bact spliceosome, PDB ID: 5Z56). SF3B1 is represented as a beige ribbon. Blue spheres: NDD-associated variants, orange spheres: frequent cancer-associated variants (R625, K666 and K700). **E** Spatial environment of selected residues. Side chains are shown as sticks. The bottom row represents a simulation of the substituted residues (Pymol). Colours are defined as in panel D. Dotted lines indicate distances <3.5 Å. T703P: the proline chain is predicted to be closer to the Q699 and K700 residues, potentially increasing intramolecular interactions and rigidifying the structure locally; P780L may increase the number of interaction points between SF3B1 and the pre-mRNA; for N829S and R939C, each substitution shortens the side chain, potentially disrupting SF3B1 intramolecular interactions with K786 and E782 for N829S, and with Q900, F901 and V947 for R939C.

unable to restore cell proliferation, as observed for the inactive splicing isoform used as a control (SF3B1ins), expressing each of the three missense variants E722K, P780L and N829S restored the cell proliferation at a level similar to the wild type counterpart (Fig. 3C, D), as previously reported for K700E[32]. These results confirm that E722K, P780L and N829S variants do not abolish the function of the SF3B1 protein.

To investigate whether the missense variants would share similarities with cancer-associated *SF3B1* variants, we studied specific splicing events in *TMEM14C, ENOSF1, SEPT6, SF3B1*(ins), *MAP3K7, ZDHCC16* and *DLST*, which are known to be mis-spliced upon expression of the K700E variant. Using this splicing assay, we showed that only P780L, and, to a lesser extent, T703P, shared some similarities with the cancer-associated K700E variant, with lower inclusion levels (Fig. 3E).

Overall, none of the eleven NDD-associated *SF3B1* missense variants that have been studied caused a loss-of-function of the protein, suggesting a molecular mechanism different from that observed with disruptive variants (pLoF). Furthermore, two missense variants shared partial similarities with K700E, suggesting a reduced fidelity in the recognition of BP sequences.

## De novo SF3B1 missense variants E722K, P780L and N829S alter RNA splicing more moderately than K700E

To investigate whether the missense variants would affect canonical and alternative splicing at the transcriptome level, we performed RNA sequencing in K562 cells stably expressing selected missense variants (E722K, P780L and N829S), in comparison to wild type SF3B1 (Fig. 4A). We also included the most studied cancer-associated variant, K700E, as a comparison. Given that none of these variants led to a loss-of-function, we decided to induce the expression of the variants in a background in which endogenous *SF3B1* was not silenced, so that we could maintain equivalent levels of endogenous and recombinant SF3B1 protein, mimicking a heterozygous state, which was verified by western-blotting (Supplementary Fig. 6A). Moreover, at both RNA and protein levels, the three recombinant NDD-associated *SF3B1* variants and the wild type (recombinant) counterpart were expressed at a similar level (Supplementary Fig. 6B).

The alternative splicing analysis consisted of the detection and the quantification of the five classical splicing events (SE = skipped exons, A5'SS = alternative 5′ splice sites, A3'SS= alternative 3′ splice sites, RI= retained introns, MXE= mutually exclusive exon usage) using the software rMATS (turbo v4.1.2). We selected splicing events with a minimum percent spliced In (PSI) of 0.1, i.e. alternative or aberrant transcripts representing at least 10% of total transcripts at a specific junction. We analysed differentially spliced junctions associated with each variant in comparison to the wild type condition, focusing on the events with an absolute value of deltaPSI (PSI SF3B1$^{var}$− PSI SF3B1$^{wt}$) set to a minimum of 0.1. The proportion of splicing types affected by K700E was similar to that reported in the literature[36] with an edited K562-K700E cell line, validating our inducible cellular model. Overall,

differential analysis of splice junctions showed an impact of each variant on global alternative splicing, affecting both known and, mainly, novel junctions (i.e. not annotated) (Fig. 4B). The effect of NDD-associated variants was quantitatively less important than for the K700E variant. The number of genes with significantly differentially spliced transcripts was between 1374 and 1715, compared to 2022 for K700E. However, many significant differentially spliced junctions had a delta PSI between 0.1 and 0.2 (Supplementary Fig. 6), which requires to nuance the interpretation of the functional impact of such events. Of note, with a delta PSI cut-off at 0.25, the number of genes with significantly differentially spliced transcripts was 448 for E722K, 532 for P780L, 417 for N829S and 715 for K700E. When considering novel junctions, NDD-associated *SF3B1* missense variants modulate all classes of alternative splicing (Fig. 4C, Supplementary Fig. 7 and 8), but mainly SE, which is the most prevalent type of alternative splicing in human cells[38], and A3'SS events. Pairwise comparisons of proportions indicate that P780L impacted (known and novel) splicing events in a similar proportion to K700E, except for MXE ($p$ = 0,0024) that was more represented in P780L. While the proportions of A5'SS and RI were unchanged by the *SF3B1* mutational status, the proportion of SE in E722K and N829S was higher than that observed in K700E, at the expense of A3'SS events. Nevertheless, the proportion of novel splicing events was overall similar whatever the variant (Fig. 4D and Supplementary Fig. 6).

To further support these findings in a more clinically relevant model, we examined alternative splicing profiles (using rMATS) in short-term cultured lymphocytes derived from two NDD patients (carrying variants p.N829S and p.P370L) in comparison to 14 controls. Remarkably, principal component analysis (PCA) showed that the two patient samples clustered together and separately from the controls, which strongly suggests a distinct splicing signature associated with N829S and P370L variants (Fig. 4G and Supplementary Fig. 9).

To investigate whether some pathways would be predominantly affected by *SF3B1* variants, we performed a Gene set enrichment analysis (GSEA) on the list of significant genes showing aberrant splicing events identified by rMATS. When considering all splicing events, the most significant terms associated with each variant included cytoplasmic translation, ribosome biogenesis, RNA splicing and RNA processing, Nonsense Mediated Decay, DNA repair as well as axon guidance and nervous system development (Table 2, Fig. 4F and Supplementary Fig. 10). The latter term covered 59 differentially spliced genes for E722K, 60 for N829S and 70 for P780L (98 for K700E), among which 19 to 27 were A3'SS events. Moreover, the term « Disorders of nervous system development » was enriched for P780L, with misssplicing of *SIN3A, TBL1XR1, HDAC3, NCOR2* and *TBL1X*, some of which in common with N829S and E722K. When performing GSEA with KEGG pathways, the terms Ribosome, Ubiquitin mediated proteolysis and Nucleocytoplasmic transport were shared between two or three NDD-associated variants.

To further investigate the impact of each variant on the transcriptome, we performed a differential gene expression analysis

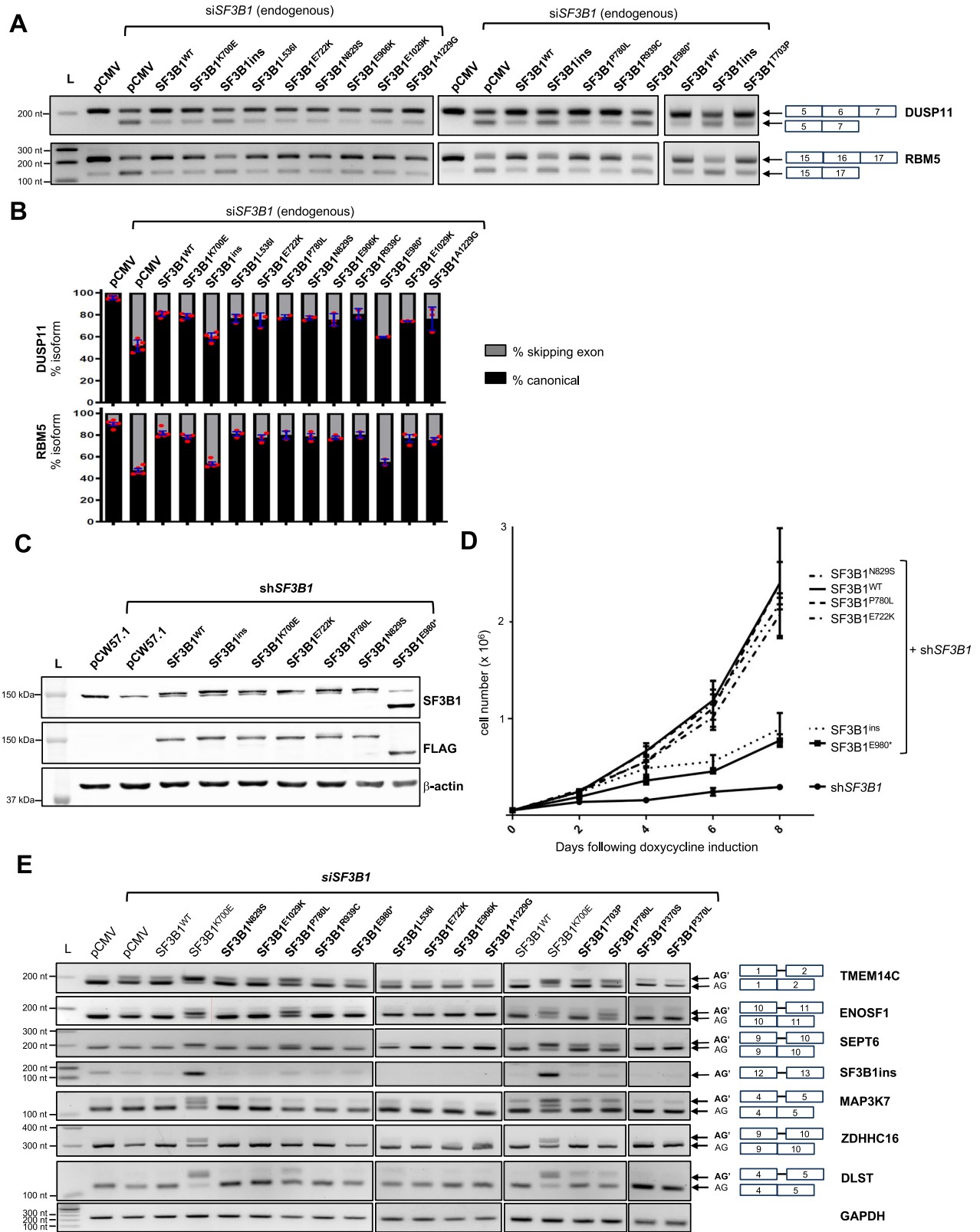

(Supplementary Data 2). The number of Differentially Expressed Genes (DEG) was low in comparison to that reported for K700E ($n = 104$), especially for N829S ($n = 4$) and E722K ($n = 5$), suggesting a more moderate overall impact on gene expression for the latter. In contrast, P780L presented with 23 DEG. Interestingly, the gene encoding the ribosomal protein RPS15A, a component of the 40S subunit which has been associated with Diamond-Blackfan anaemia[39] and several

cancers, was underexpressed in both N829S and P780L expression conditions. *MTRNR2L12*, encoding a little-studied humanin-like peptide which might have a neuroprotective role[40], was underexpressed in both E722K and N829S expressing cells. The number of DE transcripts was higher ($n = 131$ for E722K, $n = 236$ for N829S, $n = 382$ for P780L and $n = 329$ for K700E) and some of them were also misspliced (Supplementary Fig. 11). GSEA shows enrichment in genes involved in several

**Fig. 3 | Functional characterization of *SF3B1* missense variants in K562 cells.**
**A** RT-PCR analysis of exon skipping of *DUSP11* and *RBM5*, used as indicators of *SF3B1* loss-of-function, in K562 cells co-expressing si*SF3B1* and various NDD-associated variants. The splicing deficient SF3B1 "ins" isoform[32] was used as a control. **B** Digital quantification of exon skipping in *DUSP11* and *RBM5* (n = 3). **C** Steady-state SF3B1 protein levels detected by Western-Blot in K562 cells stably expressing four NDD variants under the control of a doxycycline-inducible promoter, in combination with endogenous *SF3B1* silencing (shSF3B1). Total SF3B1 proteins (endogenous and recombinant) were detected using anti-SF3B1 antibody. Recombinant SF3B1 was detected using anti-FLAG antibody. **D** Proliferation curves

of inducible K562 cells (as in **D**) following induction by doxycycline (2 microg/mL) (n = 4 biological replicates for all conditions, except for sh*SF3B1* alone for which n = 2). A two-sided Mann–Whitney test was applied and showed a significant difference (p value < 0,05) between K562 cells expressing WT, N829S, E722K or P780L variants and K562 cells expressing E980* variant (p value = 0,0286) or the inactive SF3B1ins splicing isoform (p value = 0,0286). Data are presented as mean values ± SD. **E** RT-PCR detection of aberrant transcripts known to be specifically produced upon expression of somatic *SF3B1* mutations (K700E), in K562 cells transiently expressing *SF3B1* variants of interest. This experiment was repeated three times independently with similar results. In (**B** and **D**), n refers to biological replicates.

pathways, including the regulation of transcription by RNA polymerase II, DNA metabolic process, chromosome organisation and RNA splicing, as well as in translation (initiation, regulation)(Supplementary Data 3). Given that more than 1500 NDD-associated genes have been reported so far, it is not surprising that DE transcripts and/or differentially spliced genes in our study include many genes that are known to be associated with NDD (Supplementary Fig. 12). Considering RNA splicing is tightly regulated in a cell-type-specific manner, and that splicing outcomes observed in one cellular context may not fully recapitulate those in other tissues, further validation in additional cell models would be necessary to characterize the tissue-specific effects of the variants.

### Impact of de novo *SF3B1* missense variants on the selection of cryptic 3' splice sites
To further characterize the impact of each variant on RNA splicing, we investigated whether the different variants affected the same junctions, focusing on SE and A3'SS events (Fig. 4E and Supplementary Fig. 8). For each variant, 25–32% of SE events were found in common with K700E, suggesting limited common mechanisms between somatic and germline variants. Among those, 50 SE events were shared by all four variants, while NDD-associated *SF3B1* variants shared 84 SE events (Fig. 4E). Remarkably, approximately one third of differentially spliced SE events (31% to 37.5%) was shared with at least another *SF3B1* NDD-associated variant. Nevertheless, most differentially SE were actually unique or shared with another variant.

Regarding A3'SS events, between 61% and 74% of events were variant specific. Nevertheless, 164 A3'SS events were found in common between 2 or 3 NDD-associated *SF3B1* variants (Fig. 5A and Supplementary Fig. 8). In particular, the P780L variant shared the most A3'SS events with the K700E cancer variant, with a total of n = 158, compared to n = 89 for E722K and n = 93 for N829S. As illustrated in the Sashimi plots (Supplementary Fig. 13), P780L altered the splicing of *SEPT6* less drastically than K700E, as observed by end-point RT-PCR (Fig. 3E). Other interesting examples of transcripts showing A3'SS events exclusively within both P780L and K700E variants include *ENOX2*, *HSPBP1* and the oncogenic long non-coding RNA *CRNDE*, as confirmed by an alternative method (Fig. 5B). Among the A3'SS events that were found in common (38 events, including 24 shared with K700E), we noticed a low and variable PSI, suggesting that each variant affects splicing differently in a subtle way (Supplementary Fig. 8). Importantly, when we extended the study of *CRNDE*, *ENOX2* and *HSPBP1* missplicing to the other variants, we found that L536I and T703P led to similar missplicing (Fig. 5D and Supplementary Fig. 5E).

Given the known function of SF3B1, we specifically analysed the sequence of the A3'SS junctions that were differentially used by SF3B1 variants. Remarkably, the distance between alternative AG' and canonical AG was different (Fig. 5E). For K700E, the cryptic site was on average located at a distance of 17 nucleotides from the canonical AG, as previously reported in the literature[41]. Similar results were obtained with the somatic R625H variant, validating the inducible model used in this study for splicing analysis. In contrast, the NDD-associated SF3B1 variants preferentially selected cryptic AG' that were mainly located closer to the canonical AG, with a peak centred to 7 nucleotides for

both P780L and E722K (Fig. 5E), suggesting different sequence requirements. Moreover, we found that each of the NDD-associated SF3B1 variants mostly leads to the selection of cryptic AG' that are located downstream of the canonical AG, in contrast to K700E and R625H (Fig. 5F). Notably, a second peak was visible for P780L, which mapped to the main peak observed for K700E/R625H, reflecting A3'SS events with cryptic AG' located upstream of canonical AG. Analysing the motif frequency plots for aberrant AG' and canonical AG allowed us to define sequence motifs that appeared to be relatively distinct from the one observed upon K700E expression (Fig. 5G). Nevertheless, differences between density plots were modest, suggesting that small differences in polypyrimidine tract composition rather than major branchpoint changes are likely responsible for the observed splicing differences.

Alternative splicing choices are determined by interactions between RNA-binding proteins (RBP) and *cis*-regulatory binding motifs located in introns and exons. We performed motif analyses of RBP in the vicinity of the junctions that were mis-spliced upon expression of NDD-associated SF3B1 variants, using rMAPS2[42] (Supplementary Table 5). Regarding A3'SS events, we report an enrichment in SRSF1 binding motifs and a depletion of motifs for SRSF9, RBM47, SRp40, SRp20 and PCBP2. Sequences in the vicinity of alternative SE events are enriched with motifs for the following RBP: HuR, RBM42, HNRNPA1, PCBP1 and PTBP1 (5' of exons), as well as SRSF1, PABPC3 and A1CF (3' of exons), and are depleted of motifs for RBMS3 and CNOT4 (5' of exons), as well as for HNRNPA1L2 and RBMS3 (5' of exons). This analysis suggests that SF3B1 variants may favour usage of cryptic splice sites in cooperation with specific RBP.

## Discussion
*SF3B1* is the splicing factor gene most frequently mutated in cancer, yet no germline variant has been reported. Here, we report a series of 26 cases establishing that constitutional heterozygous *SF3B1* variants impair neurodevelopment. Spliceosomopathies, caused by pathogenic variants in genes involved in the splicing machinery[3,43], often present with microcephaly, seizures or feeding difficulties, similar to our cohort. Our series includes individuals with predicted loss-of-function (pLoF) variants (i.e. premature stop codon, frameshift, canonical splice sites) (n = 9) and missense variants (n = 17). Although a larger cohort would be required to clearly establish a genotype-phenotype correlation, clinical data suggest that patients carrying *SF3B1* missense variants generally exhibit a more severe and syndromic phenotype, with congenital microcephaly, cardiac defects, feeding difficulties or hypotonia. Notably, the only variant inherited from an asymptomatic parent in our cohort was a pLoF, suggesting incomplete penetrance or variable expressivity, which is not uncommon in autosomal dominant NDD[44]. This dichotomy, where missense variants are associated with a more severe phenotype, has already been described in dominant and recessive disorders[45,46], and is explained by distinct molecular effects of these variants.

We identified 25 *SF3B1* variants, including 9 pLoF and 16 missense variants, two of them with recurrent positions (p.P370S/L and p.R939C). In silico analysis predicted most variants to be deleterious. While variants were distributed across the gene, missense variants

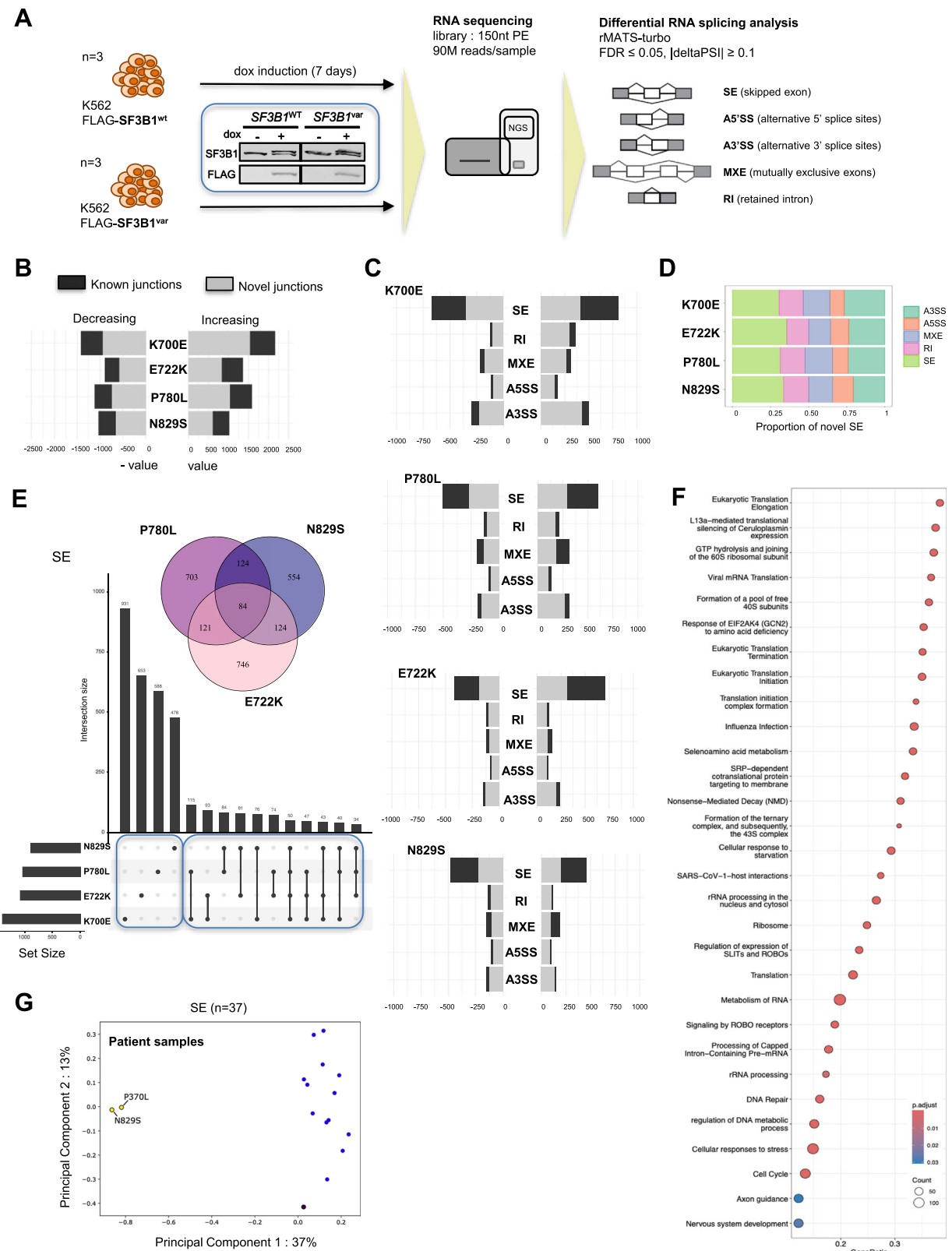

primarily clustered within the HEAT domain, except for three variants (p.P370S/L and p.R397H) affecting less structured regions. Functional complementation assays in two different cellular models showed that the eleven de novo *SF3B1* missense variants tested do not cause a loss-of-function of the protein. Targeted and genome-wide analysis of transcripts revealed that they affect RNA splicing (mostly exon skipping and A3'SS events) more modestly than the p.K700E variant, and

subtly modify the splicing of many transcripts, some of which are involved in neurodevelopmental disorders. Remarkably, GSEA analysis revealed an enrichment in genes involved in RNA metabolism, translation and ribosome biogenesis.

Thus, we can hypothesise at least two distinct molecular mechanisms: (i) a *loss-of-function* mechanism for splicing, nonsense and frameshift variants, for which SF3B1 transcripts are expected to be

**Fig. 4 | De novo *SF3B1* variants E722K, P780L and N829S alter RNA splicing.**
**A** Schematic representation of the experimental strategy used to characterise the impact of *SF3B1* variants on alternative splicing. **B** Total number of significant differentially spliced junctions associated with *SF3B1* variants, in comparison to the wild type condition. Dark and white bars indicate known junctions and novel (i.e. not annotated) junctions, respectively. The number of differentially spliced junctions was 2244 (1396 novel) for E722K, 2691 (1767 novel) for P780L, 2042 (1261 novel) for N829S and 3573 (2468 novel) for K700E when using delta PSI > 0.1.
**C** Type and number of splicing events significantly associated with each *SF3B1* variant, in comparison to the wild type condition. **D** Proportion of novel splicing events associated with each *SF3B1* variant. **E** Venn diagram and Upset Plots visualising the number of differentially spliced junctions that are in common between NDD-associated *SF3B1* variants (Venn) and K700E (Upset Plots) for Skipping Exons

(SE). Number of SE events in common with K700E: 259/1018 for E722K, 284/989 for P780L and 269/840 for N829S. Number of SE events that are variant-specific: 374/500 for P780L, 258/382 for E722K and 183/298 for N829S. **F** Dot-plot highlighting the top30 enriched REACTOME terms for differentially spliced genes (adjusted *p* value < 0.05) that were found in common between NDD-associated SF3B1 variants. Enrichment analysis was performed using a modified Fisher's exact test to evaluate term overrepresentation. Only P780L data is shown. **G** Principal component analysis (PCA) performed on PSI values of significant SE events from short-term cultured lymphocytes. RNA-Seq were performed on P1 and P26 individuals, harbouring the variants p.N829S and p.P370L, respectively, and on 14 NDD individuals without *SF3B1* variant. PCA were performed using the PSI values of significant events for alternative exon skipping (*n* = 37)(see Supplementary Fig. 8 for all splicing events). Blue: controls, yellow: *SF3B1* individuals.

**Table 2 | The TOP30 enriched REAC terms found in common, for differentially spliced genes**

| Term name | Term size | P780L | | | E722K | | | N829S | | | K700E | |
| --- | --- | --- | --- | --- | --- | --- | --- | --- | --- | --- | --- | --- |
| | | intersec. size | *p* value | | intersec. size | *p* value | | intersec. size | *p* value | | Intersec. size | *p* value |
| Metabolism of RNA | 701 | 139 | 1,8E−18 | | 106 | 1,1E−12 | | 90 | 4,5E−07 | | 156 | 5,4E−20 |
| Influenza Infection | 152 | 51 | 3,1E−15 | | 43 | 1,4E−13 | | 34 | 1,2E−07 | | 53 | 4,9E−15 |
| L13a-mediated translational silencing of Ceruloplasmin expression | 112 | 42 | 1,5E−14 | | 37 | 1,3E−13 | | 29 | 9,0E−08 | | 51 | 9,6E−20 |
| GTP hydrolysis and joining of the 60S ribosomal subunit | 113 | 42 | 1,7E−14 | | 36 | 4,8E−13 | | 29 | 9,0E−08 | | 51 | 1,3E−19 |
| Eukaryotic Translation Initiation | 120 | 42 | 1,5E−13 | | 38 | 1,3E−13 | | 30 | 9,0E−08 | | 53 | 9,6E−20 |
| Eukaryotic Translation Elongation | 94 | 36 | 6,2E−13 | | 35 | 2,0E−14 | | 25 | 4,5E−07 | | 45 | 1,3E−18 |
| Formation of a pool of free 40S subunits | 102 | 37 | 1,6E−12 | | 34 | 5,6E−13 | | 26 | 4,9E−07 | | 46 | 5,6E−18 |
| Selenoamino acid metabolism | 117 | 39 | 6,7E−12 | | 33 | 1,3E−10 | | 25 | 1,9E−05 | | 48 | 7,6E−17 |
| Response of EIF2AK4 (GCN2) to amino acid deficiency | 102 | 36 | 7,4E−12 | | 32 | 1,3E−11 | | 24 | 5,6E−06 | | 46 | 5,6E−18 |
| Cellular response to starvation | 157 | 46 | 7,4E−12 | | 39 | 1,2E−10 | | 29 | 5,1E−05 | | 58 | 9,5E−18 |
| rRNA processing in the nucleus and cytosol | 192 | 51 | 1,7E−11 | | 44 | 9,2E−11 | | 32 | 1,3E−04 | | 66 | 3,3E−18 |
| Viral mRNA Translation | 90 | 33 | 1,8E−11 | | 31 | 2,4E−12 | | 22 | 9,1E−06 | | 42 | 3,6E−17 |
| Translation | 292 | 62 | 5,6E−11 | | 66 | 6,8E−15 | | 48 | 1,3E−06 | | 78 | 1,1E−14 |
| Eukaryotic Translation Termination | 94 | 33 | 6,4E−11 | | 31 | 7,3E−12 | | 23 | 5,2E−06 | | 42 | 2,5E−16 |
| rRNA processing | 202 | 51 | 1,1E−10 | | 45 | 1,3E−10 | | 32 | 3,7E−04 | | 67 | 1,0E−17 |
| SRP-dependent cotranslational protein targeting to membrane | 113 | 36 | 1,5E−10 | | 35 | 2,1E−12 | | 22 | 3,7E−04 | | 44 | 1,7E−14 |
| Nonsense Mediated Decay (NMD) | 116 | 36 | 3,3E−10 | | 35 | 4,2E−12 | | 27 | 1,4E−06 | | 50 | 2,2E−18 |
| Cellular responses to stress | 780 | 116 | 1,8E−07 | | 91 | 2,7E−05 | | 83 | 1,4E−03 | | 145 | 1,8E−12 |
| Ribosome | 153 | 38 | 2,1E−07 | | 37 | 1,8E−09 | | 26 | 2,2E−03 | | 49 | 6,3E−12 |
| Regulation of expression of SLITs and ROBOs | 171 | 40 | 2,7E−07 | | 39 | 1,7E−09 | | 32 | 1,2E−05 | | 58 | 7,5E−16 |
| Translation initiation complex formation | 59 | 20 | 2,8E−06 | | 18 | 3,1E−06 | | 14 | 1,5E−03 | | 24 | 3,0E−08 |
| Activation of the mRNA upon binding of the cap-binding complex and eIFs | 60 | 20 | 3,8E−06 | | 18 | 4,0E−06 | | 14 | 1,8E−03 | | 24 | 4,4E−08 |
| SARS-CoV-1-host interactions | 95 | 26 | 4,1E−06 | | 22 | 2,1E−05 | | 16 | 1,9E−02 | | 25 | 1,6E−04 |
| Processing of Capped Intron-Containing Pr.10-mRNA | 276 | 49 | 4,1E−05 | | 32 | 4,7E−02 | | 35 | 1,0E−02 | | 55 | 1,6E−05 |
| Signaling by ROBO receptors | 217 | 41 | 6,0E−05 | | 43 | 1,9E−08 | | 35 | 1,0E−04 | | 60 | 6,5E−12 |
| Formation of the ternary complex, and subsequently, the 43S complex | 52 | 16 | 1,8E−04 | | 15 | 6,6E−05 | | 12 | 5,8E−03 | | 21 | 3,8E−07 |
| DNA Repair | 329 | 53 | 2,5E−04 | | 42 | 3,3E−03 | | 43 | 1,4E−03 | | 61 | 4,8E−05 |
| Cell Cycle | 677 | 91 | 4,3E−04 | | 71 | 8,4E−03 | | 80 | 5,1E−05 | | 105 | 8,8E−05 |
| Nervous system development | 574 | 70 | 2,7E−02 | | 59 | 3,0E−02 | | 60 | 1,7E−02 | | 98 | 2,5E−06 |
| Axon guidance | 549 | 67 | 3,1E−02 | | 57 | 3,0E−02 | | 58 | 1,7E−02 | | 94 | 4,0E−06 |

Pathway enrichment analysis was performed on the list of differentially spliced genes using Reactome terms (with FDR < 0.05) and excluding those with more than 800 items (Fisher's exact test using R package gprofiler2). Only the terms shared between E722K, P780L and N829S variants are presented, and the ranking is based on FDR value for P780L.

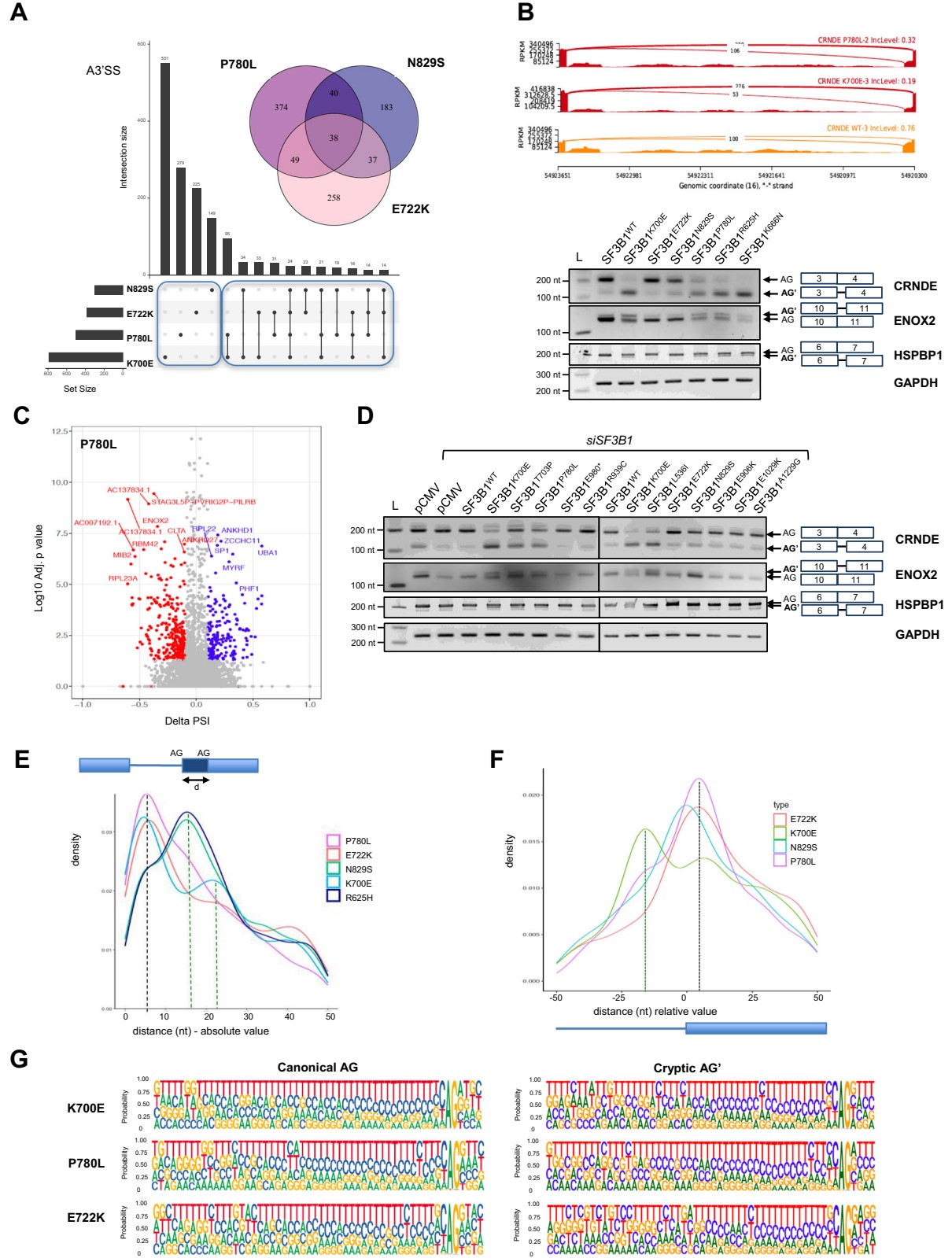

recognized and degraded by NMD, suggesting haploinsufficiency as the likely mechanism; and (ii) a *change-of-function* mechanism, leading to a remodelling of RNA splicing. Alterations of RNA splicing seem to occur at junctions that are only partially shared with those affected by K700E substitution. Indeed, the three missense variants included in the transcriptomic analysis mostly affected specific junctions, some of which overlap. Therefore, considering the heterozygous nature of

*SF3B1* variants, LoF variants may decrease overall protein levels, whereas missense variants are likely to compete with the wild type protein for incorporation into U2 snRNP particles, resulting in altered splice site recognition.

In line with the pathogenicity predictions, the p.N829S variant (patient 1) had the mildest effect on pan-transcriptomic alternative splicing, especially for A3'SS events. Still, cultured lymphocytes from

**Fig. 5 | Impact of de novo *SF3B1* missense variants on the selection of cryptic 3′ splice sites. A** Venn diagram and Upset Plots visualising the number of differentially spliced junctions in common between NDD-associated *SF3B1* variants (Venn) and K700E (Upset Plots) for A3′SS events .**B** Top: Sashimi plots representing read coverage in RPKM for A3′SS events in CRNDE in K562 cells with inducible expression of *SF3B1* variants (WT, P780L and K700E). Bottom: Validation of A3′SS events in CRNDE, ENOX2 and HSPBP1 by RT-PCR using primers allowing the amplification of both canonical and aberrant transcripts. This experiment was repeated three times independently with similar results. **C** Volcano plots showing the significant differential A3′SS splicing events upon expression of P780L variant in comparison to the wild type counterpart (see Supplementary Fig. 7 for the other variants). Differential expression was assessed with the Wald test, and *p* values were adjusted

for multiple comparisons using the Benjamini–Hochberg method (R package DESeq2). Events with an adjusted *p* value < 0.05 were considered significantly differentially expressed. **D** Effect of NDD-associated variants on the splicing of CRNDE, ENOX2 and HSPBP1 in K562 (at 48 h post-transfection). This experiment was repeated at least two times independently with similar results. **E** Density plots showing the distance (absolute value) between the cryptic/alternative AG′ and the canonical AG in K562 cells with inducible expression of E722K, P780L, N829S, K700E and R625H SF3B1 variants. **F** Density plot of the relative positions of cryptic AG′ compared to their canonical splice sites. **G** Motif frequency plots for aberrant AG′ and canonical AG for 3′ SS. The motifs are given 50 nt upstream the AG and 3 nt downstream.

this patient present a splicing signature that overall differs from control individuals. This patient, who presented a less syndromic phenotype than other patients, also harbours a de novo heterozygous missense variant in *LARP1*, which encodes an RBP that post-transcriptionally regulates the stability and translation of mRNAs and has recently been described in an autosomal dominant form of NDD[47]. While the functional impact of this *LARP1* variant is unknown, it may contribute to the patient's phenotype.

The P780L variant shared the most splicing events with K700E, including well-known targets such as *SEPT6* and *CRNDE*. Extending the targeted study to more events (detected by RNAseq) implicated L536I and T703P variants in missplicing of *CRNDE, HSPBP1* and *ENOX2*. Importantly, recent studies highlight the variable effect of *SF3B1* mutations in cancer, especially in MDS. For instance, the E592K and K666N hotspots, associated with higher risk MDS, each induce a unique RNA missplicing pattern[11,12]. This suggests that each mutation may confer a unique splicing signature that influences disease phenotype. A similar scenario may occur for the NDD-associated SF3B1 variants, which might contribute to the phenotypic differences observed in patients.

RNAseq analysis suggests that P780L, E722K and N829S variants reduce the fidelity of BP recognition to a lesser extent than K700E. Remarkably, we found that the A3′SS favoured by those variants displayed sequence determinants that differed from K700E, with preferential use of cryptic AG′ located closer to the canonical AG. Such a distinctive phenotype mirrors that of non H4-H8 hot spot *SF3B1* variants, such as the bladder carcinoma-associated E902K[35] variant, which also favoured the selection of AG′ located downstream of the canonical AG.

In silico structural visualization of the substituted residues suggests possible modification of interactions between key structural residues, between SF3B1 and the mRNA (for P780L, E906K and E1029K), or with protein partners (RBMX2 for P780L). Remarkably, half of these missense variants affect residues proximal to SF3B1 interaction sites with DDX42 or/and DDX46 (PRP5) helicases[30,31], suggesting potential impairment of these interactions. This could induce local structural rearrangements, leading to BP misrecognition and cryptic or canonical AG misselection. Despite extensive studies on cancer-associated *SF3B1* hotspots, the precise molecular mechanisms underlying their effects on splicing remain elusive.

Genes associated with both cancerous (via somatic variants) and neurodevelopmental (via germline variants) conditions have been described, suggesting that factors like cell type, timing of activation (early in development or sporadic emergence) or even type of variant[48] may influence phenotypic outcomes. SF3B1 joins the short list of spliceosomal components with dual phenotypes, together with PUF60, U2AF2 and PRPF19[20,23]. Remarkably, two components of the SF3b complex, namely SF3B2 and SF3B4[49,50], are involved in both cancer and craniofacial microsomia. Notably, germline variants of *SF3B1* induce milder splicing alterations than K700E or R625H. This is consistent with the fact that extensive splicing alterations in developing tissues would be excessively detrimental, and thus not viable. In

contrast, cancer cells carrying splicing gene mutations may acquire mechanisms to tolerate the stress induced by major splicing deregulations.

Overall, our study establishes *SF3B1* as a NDD-causative gene, with direct implications for the diagnosis and genetic counselling of NDD patients, potentially ending the diagnostic odyssey. Dissecting the molecular consequences of NDD-associated *SF3B1* variants will be essential for elucidating their role in pathology and clarifying the determinant role of SF3B1 in RNA splicing and in U2 snRNP-independent processes. This study paves the way for a fascinating research area that aims to investigate the differential impact of *SF3B1* variants in the pathophysiology of cancers and NDD, with expected repercussions in both fields.

## Methods

### Patient recruitment and clinical data analysis

A total of 26 individuals were included in our study. Individual 1's variant was identified by whole exome sequencing in the framework of the HUGODIMS consortium, whose goal was to investigate the molecular basis of developmental disorders in a research setting. The 25 remaining affected individuals were enroled after submitting the *SF3B1* variant found in individual 1 to the GeneMatcher platform[51]. The variants in these 26 individuals were identified either by trio or solo exome sequencing (ES) conducted primarily in a diagnostic setting, or within various research cohorts. All data included in this study were collected and shared in compliance with institutional policies and applicable regulations. This study was approved by the ethics committee of Nantes CHU Hospital (comité consultatif sur le traitement de l'information en matière de recherche no. 14.556). All families gave written informed consent for inclusion in the study and for publication according to their respective institutional IRB approved protocols. All patients or their guardians have seen the data within the context of the publication. The authors affirm that human research participants (P1, P6, P7, P12, P16, P18, P21, P23, P25 and P26) provided written informed consent for publication of the images in Fig. 1. The individuals or their guardians have seen the photographs within the context of the publication. Clinical data were collected from the referring physician using an anonymized Excel sheet. Categorical data for 15 clinical features from all patients were converted to 0 (absent), 0.5 (present, mild), or 1 (present). Hierarchical clustering was performed using pheatmap R package, using ward.D2 clustering method keeping missing values. In this manuscript, the term predicted loss-of-function (pLOF) is restricted to nonsense, frameshift and canonical splice-site variants. Missense variants are gathered in a unique category, regardless of their functional impact.

### Prediction software and variant classification

The prediction of missense variants' pathogenicity was assessed using common in silico predictors, such as SIFT[52], PolyPhen2[53], MutationTaster[54], MetaSVM[55], CADD[56], REVEL[57] and Alphamissense[58]. CADD scores were compared with scores of rare variants (i.e. allele frequency $<10^{-5}$) in gnomAD V4.1. The mean scores were compared

using independent *t* tests. Protein missense tolerance ratio (MTR) was visualised using MTR-Viewer[59]. We adopted the 31-codon sliding window and used exome-sequencing standing-variation data in the gnomAD database, version 2.0. MTR data were downloaded from Missense Tolerance Ratio (MTR) Gene Viewer (http://mtr-viewer.mdhs.unimelb.edu.au/mtr-viewer/). A simulation of variant classification according to the American College of Medical Genetics and ClinGen recommendations[60–62] was made, with the assumption of a definitive gene-disease association. The Pymol Software (The PyMOL Molecular Graphics System, Version 2.5 Schrödinger, LLC) was used to visualise and explore structures containing SF3B1 (PDB:5Z56). Pymol's mutagenesis tool was used to produce the 2E panel.

## Cloning and directed mutagenesis

Plasmids encoding P370S, P370L, L536I, T703P, E722K, P780L, N829S, E906K, R939C, E980*, E1029K and A1229G variants of SF3B1 were produced by site directed mutagenesis (QuikChange II XL Site-Directed Mutagenesis Kit, Agilent) using pCMV-3tag-SF3B1 plasmid as a template, which contains on optimised version of SF3B1[7,32]. The correct introduction of mutations was confirmed by Sanger sequencing. Four SF3B1 variants (E722K, P780L, N829S, E980*) were introduced in the lentiviral plasmid pCW57.1 using In-Fusion® HD Cloning Plus kit (Takara), according to the manufacturer's instructions. Primers used for cloning and directed mutagenesis are listed in Supplementary Data 4. sh*SF3B1* and si*SF3B1* targeting endogenous SF3B1 mRNA (3′UTR) were designed as described previously[32].

## Generation of *SF3B1* inducible K562 cell lines

Lentiviral particles (pCW57.1) were produced by the Vect'UB vectorology core facility (Bordeaux, France). K562 cells were transduced with a multiplicity of infection of 2. Selection of stable positive cells was done with puromycin (Sigma) for *SF3B1* inducible cell lines or G418 (Sigma) for sh*SF3B1* inducible cell lines (pLKO-Tet-on plasmid). Transduction with the sh*SF3B1* construct was done in K562 stable cell lines expressing SF3B1$^{wt}$, SF3B1$^{K700E}$, SF3B1ins, SF3B1$^{E722K}$, SF3B1$^{P780L}$, SF3B1$^{N829S}$ and SF3B1$^{E980*}$ to obtain co-transduced stable cell lines. Expression of recombinant SF3B1 and/or sh*SF3B1* was induced by treating cells with 2 microg/mL of doxycycline (Sigma).

## Cell culture

The K562 and HEK293T cell lines were obtained from ATCC. K562 and K562-derived cell lines were cultured in RPMI 1640 medium (Gibco), supplemented with 10% foetal bovine serum (Gibco) and 2 mM of L-glutamine (Gibco), at 37 °C and 5% of CO2. K562 cells were transfected by electroporation using Cell Line Nucleofector™ Kit V (Amaxa, Lonza) according to the manufacturer's instructions. Transfections were done with 2 × 106 cells and 4 µg of plasmid and/or 30 pmol of SF3B1 siRNAs designed to specifically target the 3′UTR sequence, as described previously[32]. HEK293T cells were cultured in EMEM medium (Gibco) supplemented with 10% foetal bovine serum (Gibco) and were transfected with lipofectamine 2000 (Invitrogen) according to the manufacturer's instructions. Cellular proliferation curves represent the rate of cell growth, defined as the number of living cells at a time t relative to the initial number of cells.

## RNA extraction and RT-PCR

RNA was extracted from K562 and HEK293T cells using the Nucleospin RNA kit (Macherey-Nagel). Reverse transcription was performed with High Capacity cDNA reverse transcription kit (Applied Biosystems). PCR was performed using GoTaq® G2 DNA Polymerase kit (Promega). Primers used are listed in Supplementary Data 4. PCR products were analysed on 2% or 3% agarose gels or on 6% acrylamide gels. Quantification of DNA bands was performed with Image studio lite software.

## Western-blot

Proteins were extracted with a lysis buffer containing NP40 1%–SDS 0,1% and were quantified with the Dc protein kit (Biorad). 30 µg of total proteins were separated by SDS-polyacrylamide gel electrophoresis (SDS-PAGE) and transferred onto nitrocellulose membranes. We used the following antibodies: anti-SF3B1 (1:4000, A300-996A, Bethyl), anti-FLAG (1:5000, #F3165, Sigma), anti-actin (1:10000, #ab8226, Abcam), Donkey anti-mouse (1:5000, #926-32222, LICOR) and goat anti-rabbit (1:5000, #926-2211, LICOR). Immunofluorescence was detected using the Odyssey Infrared Imaging system (LICOR).

## RNA sequencing and Bioinformatic analysis

The expression of wild type or variant SF3B1 in K562 was induced for 7 days with 2 microg/mL of doxycycline (in RPMI, 10% FBS, 2 mM Gln 2 µg/mL) before RNA was extracted using Nucleospin RNA kit (Macherey-Nagel). Biological independent replicates (*n* = 3), which refer to independent cultures from the same inducible cell line pool, were realised for each condition. RNA integrity was monitored by Bioanalyser (Agilent), with RNA Integrity Numbers ranging from 9.1 to 9.8. Directional library preparation (rRNA removal) was chosen, and RNA sequencing was performed by Novogene on an Illumina NovaSeq platform, using a 150-bp paired-end sequencing strategy.

After sequencing, raw data were obtained in the fastq format. The software FastQC[63] was used for validating the quality of the data and Trimmomatic[64] was used for trimming of the adapter content and overrepresented sequences. The alignment of the trimmed sequences to the human transcriptome was performed with the Kallisto tool[65]. Final normalisation was performed using the Relative Log Expression (RLE) method[66] and only transcripts coding for proteins were kept. Differentially expressed transcripts were identified using an adjusted *p* value cutoff (FDR) < 0.05 and absolute values of Fold Change >1.5 with R-package DESeq2[67]. The R package gprofiler2[68] was used to perform pathway enrichment analysis (DB: GO:BP, KEGG, REAC and WP) with gene symbols as input. The following options were selected: correction_method = fdr, user_threshold = 0.05.

## Differential splicing analysis

The alternative splicing analysis consisted of the detection of splicing events (SE= skipped exons, A5'SS= alternative 5′ splice sites, A3'SS= alternative 3′ splice sites, RI= retained introns, MXE= mutually exclusive exon usage), the statistical comparison and the effect prediction using rMATS turbo v4.1.2 with the option –novelSS to enable detection of novel splice sites and Homo_sapiens.GRCh38.92.exon as annotation file. rMATS (replicate Multivariate Analysis of Transcript Splicing) is a statistical method for robust and flexible detection of differential AS from replicate RNA-Seq data. Only splicing events with PSI (Percent Spliced In) greater than 10% and adjusted p-value lower than 0.05 have been selected. Novel splice sites were included. The average number of reads covering the splice event was at least 10. UpSet was used to visualise the intersecting sets between 4 or more conditions.

## Lymphocyte cultures and RNA sequencing

RNAseq experiments were performed following a protocol similar as previously described for *RNU4-2*[69]. Briefly, Peripheral Blood Mononuclear Cells were isolated from EDTA blood and cultured in lymphocyte-stimulating medium for 48–72 h. Stranded RNA-Seq libraries were prepared from 100 ng total RNA using the SureSelect XT-HS2 kit (Human All Exon V8 capture probes; Ref: G9774C), followed by sequencing on an Illumina NextSeq 550 to obtain -25–30million paired-end reads per sample. Reads were aligned to GRCh38 with STAR v2.7.11a. For splicing analysis, we focused on 16 individuals sequenced in the same run: 2 *SF3B1* patients carrying N829S and P370L, and 14 probands with NDD without rare variants in SF3B1. rMATS-turbo v4.3.0 was used with 2 cases Vs 14 controls to detect alternative splicing events with the following parameters: -t paired –anchorLength 1

–libType fr-firststrand –novelSS –variable-read-length –allow-clipping. For each splicing category, rMATS outputs were filtered for mean coverage >10, ΔPSI > 0.05 and FDR < 0.1. We kept only one event by gene, the most significant. To visualize results, we performed principal component analysis using sklearn (v1.7.1) in python (v.3.12.2).

## Statistics

The statistical analysis was performed using R version 4.3.1. Furthermore, the prop.test function was used to test proportions of different types of events between groups.

## Reporting summary

Further information on research design is available in the Nature Portfolio Reporting Summary linked to this article.

## Data availability

The RNA sequencing data generated from patient-derived lymphocytes have been deposited in the European Genome–Phenome Archive (EGA, http://www.ebi.ac.uk/ega), under accession code EGAS50000001473, and are subjected to a data processing agreement due to their sensitive nature. Access will be provided only for health or medical or biomedical research, only for non commercial use, and a collaboration with the primary study investigator is required. Due to the sensitive nature of human genetic data, controlled access to human genome data is necessary to protect participant privacy and to ensure compliance with ethical and legal standards. Individual genome data provided within a clinical diagnostic setting, which are stored on secure hospital servers, cannot be made available due to ethical and regulatory considerations, including patient consent limitations and data protection regulations. However, whole exome sequencing data performed in the framework of the HUGODIMS consortium, as well as the exome sequencing data generated in a research setting, are available from the corresponding author under controlled access. Access will be restricted to qualified researchers for non-commercial research purposes and subject to a data processing agreement. Data access requests for RNA sequencing and exome sequencing data generated in a research setting should be submitted to the corresponding author and will be reviewed within 30 days by a data accessibility committee to ensure that data access complies with ethical and legal standards respective to the corresponding projects. The RNA sequencing data generated in this study on K562 cells have been deposited in the Gene Expression Omnibus database (https://www.ncbi.nlm.nih.gov/gds) under accession code GSE287369. All other data supporting the findings of this study are available within the paper and its supplementary information files. The source data underlying Figs. 3 and 5 and Supplementary Fig . 4-6 are provided as a Source Data file. Source data are provided with this paper.

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

## Acknowledgements

The authors wish to thank the patients and families included as part of this study. The authors acknowledge HUGODIMS (Western France exome-based trio approach project to identify genes involved in

intellectual disability); funding for HUGODIMS is supported by a grant from the French Ministry of Health and from the Health Regional Agency from Poitou-Charentes (HUGODIMS, 2013, RC14_0107). This study was funded by INSERM, by the French League against cancer (*la Ligue contre le Cancer*, committees 29 and 35), by the French Biomedecine Agency and by the association Gaetan Saleun. T.B. was funded by the Brittany Region and the Ministère de l'Enseignement Supérieur, S.Ch. and C. D. were funded by the Ministère de l'Enseignement Supérieur de la Recherche et de l'Innovation. This study was supported by the « Priority Research Programme on Rare Diseases » of the French Investments for the Future Programme, project MultiOmixCare. JRL was supported in part by US National Institutes of Health NS105078 and HG011758. D.G.C. was supported by the Child Neurologist Career Development Programme K12 and Muscular Dystrophy Association Development Grant (873841). The authors thank the vectorology core facility Vect'UB in Bordeaux for the production of lentiviral particles, and thank the Centre of Biological Resources in Brest (CHU).

## Author contributions

K.U. and T.B. contributed equally. K.U., T.B., and S.K. contributed to the design of the study; T.B., S.Ch., C.D., S.C. performed the experiments; M.P.S.-B. performed the bioinformatic analysis (RNAseq); D.G.B. and K.U. contributed to the bioinformatic analysis; T.Be., L.D.S.F and B. Co. provided data on human samples; K.U., T.B., S.K., S.Ch. and D.G.B. contributed to data interpretation; M.P., S.H., P.W., H.W., B.X., V.S., M.C., C.Z., C.C., T.P., S.G., B.T., T.C., C.P., T.H., T.R., M.M., K.I., J.L., E.Z., I.G., S.W., M.B., L.F., J.M., J.D., C.J., L.LM., H.VE., D.C., L.DW., G.B., C.Pe. and L.D. contributed to the clinical data collection; K.U., S.K., C.G., M.C., S.B., P.S., Y.X., Y.W., M.D-F., D.L., P.C., F.T-M-T., A.D-P., A.V., J. Lu., P.P-B., R.M. and D.S. contributed to the exome/genome data analysis; C.F. supported the study; K.U., S.K., T.B. and D.B wrote the manuscript; E.L., L.C. reviewed the manuscript; D.G.B. designed and supervised the research. All authors read the manuscript.

## Competing interests

The authors declare no competing interests.

## Additional information

Kevin Uguen[1,2,3,48], Tiffany Bergot[1,4,48], Marie-Pier Scott-Boyer[5], Solène Chapalain[1], Camille Desdouets[1], Séverine Commet[1], Changlian Zhu[6,7], Yiran Xu[7], Yangong Wang[8], Tony Roscioli[9,10], Frederic Tran-Mau-Them[11], Laurence Faivre[12], Julien Maraval[12], Julian Delanne[12], Anne-Sophie Denommé-Pichon[11], Antonio Vitobello[11], Céline Jost[12], Marc Planes[2,3], Susan Hiatt[13], Patricia Wheeler[14], Claudia Gonzaga-Jauregui[15], Heng Wang[16], Baozhong Xin[16], Valerie Sency[16], Michael C. Kruer[17,18], Somayeh Bakhtiari[17,18], Patrick Sulem[19], Cynthia Curry[20], Trine Prescott[21], Gertrud Strobl-Wildemann[22], Theresa Brunet[23], Martine Doco Fenzy[24,25], Thomas Courtin[26,27], Céline Poirsier[28], Trine Bjørg Hammer[29,30], Christina D. Fenger[29], Melissa MacPherson[31], Kosuke Izumi[32], Jacqueline Leonard[32], Dong Li[32], Elaine H. Zackai[32], Ian A. Glass[33], Scott Ward[34], Philippe M. Campeau[35], Maria Carla Hermida Borroto[35], Laurence Le Moigno[36], Hilde Van Esch[37], Liesbeth De Waele[38], Daniel G. Calame[39], James R. Lupski[40], Giulia Barcia[41], Cristina Peduto[41], Pauline Planté-Bordeneuve[41], Lucie Dupuis[42], Roberto Mendoza-Londono[42], Dimitri J. Stavropoulos[43], Jennifer Gillibert-Duplantier[44], Thomas Besnard[24,25], Laura Do Souto Ferreira[24], Benjamin Cogné[24,25], Stéphane Bézieau[24,25], Arnaud Droit[5,45], Laurent Corcos[1], Eric Lippert[1,46,47], Claude Férec[1], Sebastien Küry[24,25] & Delphine G. Bernard[1,47] ✉

[1]Univ Brest, Inserm, EFS, UMR 1078, GGB, Brest, France. [2]Service de Génétique Médicale, CHU de Brest, Brest, France. [3]Centre de Référence Déficience Intellectuelle et Polyhandicap de causes rares, CHU de Brest, Brest, France. [4]Currently in the Division of Structural Biology, The Institute of Cancer Research, London, England. [5]CHU de Québec–Laval University Research Center, Quebec City, QC, Canada. [6]Center for Brain Repair and Rehabilitation, Institute of Neuroscience and Physiology, University of Gothenburg, Gothenburg, Sweden. [7]Henan Key Laboratory of Child Brain Injury and Henan Pediatric Clinical Research Center, Institute of Neuroscience and Third Affiliated Hospital of Zhengzhou University, Zhengzhou, China. [8]Institutes of Biomedical Sciences and Children's Hospital, Fudan University, Shanghai, China. [9]New South Wales Health Pathology Randwick Genomics, Prince of Wales Hospital, Sydney, NSW, Australia. [10]Neuroscience Research Australia (NeuRA), University of New South Wales Sydney, Sydney, NSW, Australia. [11]Université Bourgogne Europe, CHU Dijon Bourgogne, Laboratoire de Génomique Médicale, Centre Neomics, FHU TRANSLAD, Centre de recherche Translationnelle en Médecine

moléculaire—Inserm UMR1231, équipe GAD, Dijon, France. [12]Université Bourgogne Europe, CHU Dijon Bourgogne, Inserm, CTM UMR1231, équipe GAD, FHU TRANSLAD, Centre de génétique, Centre de référence Anomalies du Développement et Syndromes Malformatifs, Centre de référence Déficiences Intellectuelles de Causes Rares, et Centre de référence GénoPsy, Dijon, France. [13]HudsonAlpha Institute for Biotechnology, Huntsville, AL, USA. [14]Division of Genetics, Arnold Palmer Hospital for Children-Orlando Health, Orlando, FL, USA. [15]International Laboratory for Human Genome Research, Laboratorio Internacional de Investigación sobre el Genoma Humano, Universidad Nacional Autónoma de México, Juriquilla, México. [16]DDC Clinic for Special Needs Children, Middlefield, OH, USA. [17]Pediatric Movement Disorders Program, Division of Pediatric Neurology, Barrow Neurological Institute, Phoenix Children's Hospital, Phoenix, AZ, USA. [18]Departments of Child Health, Neurology, Cellular & Molecular Medicine and Program in Genetics, University of Arizona College of Medicine, Phoenix, AZ, USA. [19]deCODE Genetics/Amgen, Inc, Reykjavik, Iceland. [20]Genetic Medicine, University of California, San Francisco, Fresno, CA, USA. [21]Department of Medical Genetics, Telemark Hospital Trust, Skien, Norway. [22]Department of Human Genetics, MVZ Humangenetik Ulm, Ulm, Germany. [23]Institute of Human Genetics, Klinikum Rechts der Isar, School of Medicine and Health, Technical University of Munich, Munich, Germany. [24]Nantes Université, CHU de Nantes, Service de Génétique médicale, Nantes, France. [25]Nantes Université, CHU Nantes, CNRS, INSERM, l'institut du thorax, Nantes, France. [26]Sorbonne Université, Institut du Cerveau—Paris Brain Institute—ICM, Inserm, CNRS, Paris, France. [27]Assistance Publique Hôpitaux de Paris, Hôpital Pitié-Salpêtrière, Département de Génétique, DMU BioGeM, Paris, France. [28]Département de génétique médicale, CHU Reims, Reims, France. [29]Department of Epilepsy Genetics and Personalized Treatment, The Filadelfia Danish Epilepsy Centre, Dianalund, Denmark. [30]Department of Clinical Genetics, Copenhagen University Hospital, Copenhagen, Denmark. [31]Department of Medical Genetics, Faculty of Medicine and Dentistry, University of Alberta, Alberta Health Services, Edmonton, AB, Canada. [32]Division of Human Genetics, Children's Hospital of Philadelphia, Philadelphia, PA, USA. [33]Division of Genetic Medicine, Department of Pediatrics, University of Washington, Seattle, WA, USA. [34]Department of Pediatrics, Division of Medical Genetics and Genomic Medicine, Vanderbilt University Medical Center, Nashville, TN, USA. [35]Centre de Recherche Azrieli du CHU Sainte-Justine, University of Montreal, Montreal, QC, Canada. [36]Service de Pédiatrie et Unité d'Urgence Pédiatrique, Centre Hospitalier de Cornouaille, Quimper, France. [37]Center for Human Genetics, University Hospitals Leuven, Herestraat 49, Leuven, Belgium. [38]Department of Child Neurology, University Hospitals Leuven, Herestraat 49, Leuven, Belgium. [39]Section of Pediatric Neurology, Department of Pediatrics, Baylor College of Medicine, Houston, TX, USA. [40]Department of Molecular and Human Genetics, Baylor College of Medicine, Houston, TX, USA. [41]Service de Médecine Génomique des Maladies Rares, Hôpital Necker - Enfants Malades, Assistance Publique-Hôpitaux de Paris, Paris, France. [42]Division of Clinical and Metabolic Genetics, Department of Pediatrics, The Hospital for Sick Children and University of Toronto, Toronto, ON, Canada. [43]Department of Laboratory Medicine and Pathobiology, University of Toronto, Toronto, ON, Canada. [44]VECT'UB, TBMCore, CNRS UAR 3427, INSERM US005, Université de Bordeaux, Bordeaux, France. [45]Department of Molecular Medicine, Faculty of Medicine, Laval University, Quebec City, QC, Canada. [46]Service d'hématologie biologique, CHU de Brest, Brest, France. [47]CRB Santé du CHU de Brest, Brest, France. [48]These authors contributed equally: Kevin Uguen, Tiffany Bergot. ✉e-mail: delphine.bernard@univ-brest.fr

