## [Transparent Peer Review file · Nature Communications]

De novo variants in the splicing factor gene SF3B1 are associated with neurodevelopmental disorders.

Corresponding Author: Dr Delphine BERNARD

Version 0:

Reviewer comments:

Reviewer #1

(Remarks to the Author)

Uguen, Bernot and colleagues report a detailed analysis of the molecular effects of mutations in the splicing factor SF3B1 that they have identified in a cohort of 26 patients displaying neurodevelopmental disorders. The mutations are scattered through most of the primary amino acid sequence of the protein, are distinct from those that appear as somatic mutations in various cancers and can be classified in two main groups, those predicted to cause loss of function (e.g. in splice sites or causing stop codons) and that they turn out to be inherited, and those harboring missense mutations of less clear predicted significance. Remarkably, the latter group is associated with stronger pathological phenotypes, including beyond neurodevelopmental disorders. Functional characterization in a depletion/complementation system using cell lines shows that the novel mutations do not compromise the general activity of SF3B1 (sustaining cell proliferation) nor induce activation of cryptic 3' splice sites characteristic of cancer mutations in SF3B1. RNA-seq analyses upon ectopic expression of the mutant proteins, however, show substantial splicing changes associated with the novel missense mutations, most frequently private to individual mutants. Genes important for nervous system development are enriched among the common targets, opening the possibility that their alteration in cells expressing the mutant proteins may eventually explain at least part of the associated phenotypes.

These are very relevant observations documenting for the first time the association between neurodevelopmental disorders and mutations in SF3B1, a key core splicing factor frequently mutated in cancer and target of anti-tumor drugs. The detailed molecular characterization of the splicing activities associated with a substantial number of mutants sheds light both on common principles of SF3B1 alterations as well as on the distinct properties conferred by the different mutations and their correlation with the severity of symptoms, highlighting the variety of regulatory functions of a core component of the spliceosome in charge of the fundamental role of recognizing the branch point in pre-mRNAs.

In my opinion the following revisions could help to improve the manuscript:

1. Figure 2E: it is not easy to understand from the figure which interactions (with which residues and the predicted consequences) are affected by the indicated mutations. Additional discussions in the main text and/or Figure legend would be helpful.
2. Figure 3D, E: "SF3B1ins" is referred to in the text or in panel 3D as a LoF control, but in panel 3E is presented as a transcript species produced under expression of K700E variant. This needs to be clarified.
3. Figure 5E: it is unclear from the text or Figure legend what is the difference between the left and right panel. Nor what is the purpose of the inset "for reviewers only".
4. Figure 5G: the significance of the differences between density plots is unclear because the differences are small and affect the composition of polypyrimidine tracts, which are in general quite degenerate and unstructured. The analyses that would make a difference would be the mapping or prediction of the positions of branch points used in the absence or presence of mutations.
5. Discussion: the sentence in lines 472-474 may be confusing for some readers, because on one hand states that germline mutations display milder phenotypes and on the other points out to the wider, presumably deleterious effects of mutations in

SF3B1 that would occur during development. More generally, it may be worth explaining more clearly that, given the heterozygous nature of SF3B1 mutants, LoF mutations may “simply” reduce the total levels of the protein, while missense mutations likely compete with the wild type protein for incorporation into U2 snRNP particles and induce changes in splice site recognition that may function in a dominant negative fashion to induce phenotypes.

Juan Valcárcel

Reviewer #2

(Remarks to the Author)

Kevin Uguen and colleagues report de novo variants in SF3B1 as the genetic cause of a new, clinically variable neurodevelopmental syndrome. SF3B1 encodes the splicing factor 3b, subunit 1, a component of the U2 snRNP complex of the spliceosome, which mediates the recognition of the branch point. This patient cohort has mainly been assembled through Genematcher. Somatic recurrent missense variants of SF3B1 have been previously reported in multiple cancer types but this is the first report of constitutional variants in this gene underlying a genetic NDD. The authors then performed functional studies for 9 out of the 13 missense variants identified in a cell model that has previously been used to assess the impact of somatic variants identified in cancer. They show evidence that the variants associated with NDD have a different and overall milder impact on splicing than that previously reported for the most recurrent variant (K700E) found in cancers.

The identifications of variants in SF3B1 in NDD and the comparison of their functional impact with that of variants previously identified in cancer is both original and compelling. The methods and designed experiments are sound, and the results convincingly support the conclusions. The study is particularly timely, given the growing interest in splicing-related genes in NDD.

My main critique is linked to the description of the results and the order in which they appear in the manuscript, which makes the text sometimes difficult to follow.

To improve the clarity and readability of the manuscript, I listed specific points below:

1. Instead of “Novel”, the title could explicitly include the mode of inheritance of the variants

Introduction

2. The introduction is lengthy and currently gives the (false) impression of covering the entire field of genes involved in splicing with NDD whereas many genes are missing (to cite just a few: HNRNPU, HNRNPK; HNRNPC, HNRNPR, HNRPA2B1, SNRNP200, NOVA2, ...). I recommend streamlining the introduction to focus on concepts essential for understanding the study while avoiding broad generalities on NDDs and genes not directly related to this manuscript. For example, details on the nature and impact of SF3B1 variants in cancer would be more relevant to give some background on the findings obtained in this manuscript.

3. Page 3 line 113 and page 6, lines 255 and 258: the terms “change-of-function” and “modification-of-function” are unclear. Do the authors mean “gain-of-function”?

4. Page 3, line 122: we usually consider that NDD of genetic origin affect 3-4% of the population maximum. 10% probably includes very mild” forms and non-genetic forms of NDD.

5. Page 3, line 130: please correct “RNU-2P” to “RNU2-2P”

Results

6. page 4, lines 148-154: The authors should briefly specify how the variants were identified and, if possible, provide some context on the nature and size of the cohorts used for variant discovery.

7. Could the authors indicate for each of the 26 patients if other variants have been identified by exome or genome analysis that could account for the patients’ phenotype?

8. The clinical part lacks details about the nature of the variants associated with each feature. The authors may want to consider having the paragraph detailing the variants (lines 212-250)) before the clinical paragraphs so they can relate phenotypic features to specific variants. For instance, when discussing the three inherited cases, the exact variant details could be specified. This restructuring would also improve the clarity of genotype-phenotype correlations (Page 5, Lines 196–208).

9. It is unclear if dividing clinical features into multiple subheadings (head and neck, cardiac, etc.) is necessary. Removing the subheadings and linking these sections might improve readability and flow.

10. The abbreviation “pLoF” appears before it is defined (putative loss-of-function, Line 212). Consider defining it earlier.

11. How do the authors classify variants as putative loss-of-function(pLoF) in their genotype-phenotype correlations? Are they referring only to nonsense, canonical splice site variants (i.e., variants introducing premature termination codons) or do they also incorporate functional evidence? Additionally, please consider that missense variants can also result in loss of function(LoF), and LoF and gain of function(GoF) are not always mutually exclusive-; a single variant may exhibit both effects.

12. Page 5, Line 218: The authors should be more specific when referring to pathogenicity criteria. For example, rather than stating “pathogenicity predictors”, they should explicitly list the predictive tools used (the reader only finds this out two sentences later).

13. Page 5, lines 227-233: It is difficult to distinguish which variants are associated with cancer and which ones were identified in this study. Including an introductory paragraph earlier about somatic variants in SF3B1 found in cancer and their impact may improve clarity (see previous comment regarding the introduction).

14. Page 6, line 271: A similar issue with the skipping of exon 6 in DUSP11 and exon 11 in RBM5. Without prior context on these splicing events, the rationale is difficult to follow.

15. Page 6, line 272: Could the authors clarify whether they overexpress the SF3B1 p.Glu980* protein? If so, did they first

verify, at least theoretically, that this variant escapes nonsense-mediated decay? Otherwise, the most likely consequence of this variant (and other pLoF variants) would be haploinsufficiency (without truncated protein produced).

16. Similarly, have the authors investigated whether any of the missense variants identified in patients could impact SF3B1 splicing, such as by disrupting an ESE or creating/enhancing a novel splice site before overexpressing them?

17. Page 6, lines 272-276: The same variant is referred to using different nomenclatures (p.Glu980*/p.Glu980Ter). Please homogenize.

18. Page 6, lines 278-279: The authors suddenly shift to the one-letter amino acid code (e.g., E980*). For clarity, the variant nomenclature should be consistent throughout the text.

19. Page 12, line 566: Could the authors clarify what is meant by 'biological replicates' in the RNAseq experiments? Does this refer to independent cultures from the same immortalized cell clone (inducible cell lines) or were three different clones used? If the same clone was used, could variant-specific splicing be due to clonal effects? Please provide additional information in the methods section.

20. Could the authors provide further discussion on how variant-specific splicing might explain the phenotypic differences observed in patients?

21. While the rationale for using a cell model previously used to study splicing in cancer-specific variants is clear, could the authors discuss how the RNAseq results may be influenced by cell-type specificity?

Figures

22. Figure 1A: The heatmap is an effective way to visualize clinical features, but the meaning of the three clusters is unclear. It also seems that a significant portion of the clustering may be driven by missing data, rather than meaningful differences.

Reviewer #3

(Remarks to the Author)

In the manuscript by Uguen et al., the authors report several SF3B1 mutations found in a cohort of 26 patients with neurodevelopmental diseases (NDDs). Those mutations, mainly missense variants, are different from the hotspot mutations seen in cancer. Among the SF3B1 mutations seen in NDD patients, the authors focused on SF3B1 E722K, P780L, N829S to perform RNAseq analysis in K562 cells stably expressing these variants. The authors showed that SF3B1 E722K, P780L, N829S expression induces splicing changes compared to SF3B1 WT, to a lesser extent than the cancer associated K700E variant.

Major comments:

-The authors used K562 and HEK293T cells to study the splicing changes due to the described SF3B1 variants. These cancer cell lines are not an ideal model in this study due to the high mutational burden in these cancer cell lines and the non-physiologic SF3B1 expression. It would be ideal for the authors to use patient-derived fibroblasts, lymphoblastoid cell lines, or even primary human blood (or other cells) for the splicing analysis.

-The authors claim that SF3B1 variants described in this manuscript are associated with NDDs, but they do not include in their analysis a comparison with healthy controls, or NDD patients with SF3B1 WT. What is the frequency of the reported SF3B1 mutations in healthy controls, or NDD patients from HUGODIMS consortium?

-The three variants that the authors focused on for the RNAseq analysis are single, De Novo variants. However, in the cohort described in this study there was three inherited SF3B1 variants, namely p.(Glu809AspfsTer7), p.(Val695TrpfsTer34) and p.(Arg166Ter) and two recurrent variants, namely P370S/L and R939C. Analysis of the splicing changes due to these inherited or recurrent SF3B1 variants could have been a stronger argument that SF3B1 is involved in NDDs.

-The authors only mention in line #444 that patient 1 also harbors LARP1 mutation. A systematic description of all the genetic mutations, or absence of it, in the 26-patient cohort is important to associate SF3B1 with NDDs in this cohort.

Reviewer #4

(Remarks to the Author)

Version 1:

Reviewer comments:

Reviewer #1

(Remarks to the Author)

The authors have addressed all the points raised in my previous report and I am happy to recommend publication of the manuscript in Nature Communications.

Reviewer #2

(Remarks to the Author)

The authors have adequately addressed all points raised in the first round of review. I have no further comments.

Reviewer #3

(Remarks to the Author)

The authors have addressed my initial comments and questions. I have no further issues with the manuscript.

Reviewer #4

(Remarks to the Author)

Point-by-point response to the reviewers' comments

Reviewer #1 (Remarks to the Author):

Uguen, Bernot and colleagues report a detailed analysis of the molecular effects of mutations in the splicing factor SF3B1 that they have identified in a cohort of 26 patients displaying neurodevelopmental disorders. The mutations are scattered through most of the primary amino acid sequence of the protein, are distinct from those that appear as somatic mutations in various cancers and can be classified in two main groups, those predicted to cause loss of function (e.g. in splice sites or causing stop codons) and that they turn out to be inherited, and those harboring missense mutations of less clear predicted significance. Remarkably, the latter group is associated with stronger pathological phenotypes, including beyond neurodevelopmental disorders. Functional characterization in a depletion/complementation system using cell lines shows that the novel mutations do not compromise the general activity of SF3B1 (sustaining cell proliferation) nor induce activation of cryptic 3' splice sites characteristic of cancer mutations in SF3B1. RNA-seq analyses upon ectopic expression of the mutant proteins, however, show substantial splicing changes associated with the novel missense mutations, most frequently private to individual mutants. Genes important for nervous system development are enriched among the common targets, opening the possibility that their alteration in cells expressing the mutant proteins may eventually explain at least part of the associated phenotypes.

These are very relevant observations documenting for the first time the association between neurodevelopmental disorders and mutations in SF3B1, a key core splicing factor frequently mutated in cancer and target of anti-tumor drugs. The detailed molecular characterization of the splicing activities associated with a substantial number of mutants sheds light both on common principles of SF3B1 alterations as well as on the distinct properties conferred by the different mutations and their correlation with the severity of symptoms, highlighting the variety of regulatory functions of a core component of the spliceosome in charge of the fundamental role of recognizing the branch point in pre-mRNAs.

In my opinion the following revisions could help to improve the manuscript:

1. Figure 2E: it is not easy to understand from the figure which interactions (with which residues and the predicted consequences) are affected by the indicated mutations. Additional discussions in the main text and/or Figure legend would be helpful.

We thank the reviewer for this comment. To facilitate the reading of Figure 2E, we extended the corresponding legend and detailed the amino acids likely to be involved in the modified interactions.

2. Figure 3D, E: "SF3B1ins" is referred to in the text or in panel 3D as a LoF control, but in panel 3E is presented as a transcript species produced under expression of K700E variant. This needs to be clarified.

We thank the reviewer for pointing out this lack of clarity regarding SF3B1ins. We clarified this point both in the main text and in the legend of Figure 3.

Main text : "As a control, we included a condition in which an inactive isoform of SF3B1 was expressed, namely SF3B1ins, which we previously characterized as an aberrant transcript produced upon expression of K700E".

Legend : “Expression of the SF3B1 splicing isoform “ins”, which we reported previously as splicing deficient, was used as a control.” (panel 3A); “...K562 cells expressing the inactive SF3B1ins splicing isoform” (panel 3D).

To increase readability, we also replaced the term “WT ins” by “ins” in Figure 3.

3. Figure 5E: it is unclear from the text or Figure legend what is the difference between the left and right panel. Nor what is the purpose of the inset “for reviewers only”.

We thank the reviewer for this comment. The right panel of Figure 5E included an additional condition (cancer-associated R625H variant) that perfectly matches the K700E condition. To avoid redundancy, we deleted the left panel, as the right panel is the most comprehensive.

Inserting the panel from Liu et al (*Cancer Discovery* **10**, 806–821 2020) was meant to underline the consistency between their results and ours, regarding the distance between cryptic and canonical AG site for the cancer-associated SF3B1 variants. If the policy of Nature Communications allows such insert from another article, we would be inclined to keep it.

4. Figure 5G: the significance of the differences between density plots is unclear because the differences are small and affect the composition of polypyrimidine tracts, which are in general quite degenerate and unstructured. The analyses that would make a difference would be the mapping or prediction of the positions of branch points used in the absence or presence of mutations.

We thank the reviewer for this comment. Following your suggestion, we performed branch-point (BP) analyses using the R package branchpointer. For A3SS events showing significant differences between the variants and WT, a 100-nucleotide region upstream of the 3’ splice site (AG and AG’) was extracted and analyzed for each of the four variants using the predictBranchpoints function. Only the branchpoints with a highest prediction probability were retained for further analysis. We compared the predicted distances of branchpoints to the 3’ splice sites, either canonical (AG) or cryptic (AG’). The distance differences between the predicted branchpoint and corresponding 3’ splice site (canonical vs cryptic) overlay the density plot reported in Figure 5E (see figure below, for reviewers).

Distribution of Distance Differences Between the Predicted Branchpoint and 3’ Sites (Canonical vs. Cryptic). Density plot showing the absolute difference in predicted branchpoint-to-3’ splice site distance between cryptic and canonical splice sites across the different variants.

These prediction results indicate that, although A3SS events are altered between variants and WT, the predicted branchpoint usage and positions remain largely conserved, suggesting that small differences in polypyrimidine tract composition rather than major branchpoint changes are likely responsible for the observed splicing differences. Nonetheless, this interpretation should be tempered by the fact that BP predictions are not straightforward.

We propose to modify the main text to balance the interpretation of the motif frequency plots : “Nevertheless, differences between density plots were modest, suggesting that small differences in polypyrimidine tract composition rather than major branchpoint changes are likely responsible for the observed splicing differences.”

5. Discussion: the sentence in lines 472-474 may be confusing for some readers, because on one hand states that germline mutations display milder phenotypes and on the other points out to the wider, presumably deleterious effects of mutations in SF3B1 that would occur during development. More generally, it may be worth explaining more clearly that, given the heterozygous nature of SF3B1 mutants, LoF mutations may “simply” reduce the total levels of the protein, while missense mutations likely compete with the wild type protein for incorporation into U2 snRNP particles and induce changes in splice site recognition that may function in a dominant negative fashion to induce phenotypes.

We thank the reviewer for this suggestion. The sentence in lines 472-474 may indeed be confusing. We rephrased it. We also added a sentence in the paragraph related to the molecular mechanisms.

Juan Valcárcel

Reviewer #2 (Remarks to the Author):

Kevin Uguen and colleagues report de novo variants in SF3B1 as the genetic cause of a new, clinically variable neurodevelopmental syndrome. SF3B1 encodes the splicing factor 3b, subunit 1, a component of the U2 snRNP complex of the spliceosome, which mediates the recognition of the branch point. This patient cohort has mainly been assembled through Genematcher. Somatic recurrent missense variants of SF3B1 have been previously reported in multiple cancer types but this is the first report of constitutional variants in this gene underlying a genetic NDD. The authors then performed functional studies for 9 out of the 13 missense variants identified in a cell model that has previously been used to assess the impact of somatic variants identified in cancer. They show evidence that the variants associated with NDD have a different and overall milder impact on splicing than that previously reported for the most recurrent variant (K700E) found in cancers.

The identifications of variants in SF3B1 in NDD and the comparison of their functional impact with that of variants previously identified in cancer is both original and compelling. The methods and designed experiments are sound, and the results convincingly support the conclusions. The study is particularly timely, given the growing interest in splicing-related genes in NDD. My main critique is linked to the description of the results and the order in which they appear in the manuscript, which makes the text sometimes difficult to follow.

To improve the clarity and readability of the manuscript, I listed specific points below:

1. Instead of “Novel”, the title could explicitly include the mode of inheritance of the variants

We replaced “Novel” by “de novo”, as (i) most variants appeared de novo and (ii) our study focuses essentially on de novo variants.

Introduction

2. The introduction is lengthy and currently gives the (false) impression of covering the entire field of genes involved in splicing with NDD whereas many genes are missing (to cite just a few: HNRNPU, HNRNPK; HNRNPC, HNRNPR, HNRPA2B1, SNRNP200, NOVA2, ...). I recommend streamlining the introduction to focus on concepts essential for understanding the study while avoiding broad generalities on NDDs and genes not directly related to this manuscript. For example, details on the nature and impact of SF3B1 variants in cancer would be more relevant to give some background on the findings obtained in this manuscript.

We thank the reviewer for this recommendation. We shortened the paragraph related to NDD and focused on the core and auxiliary components of the U2 snRNP complex to which SF3B1 belongs. We also slightly extended the paragraph on somatic SF3B1 variants.

3. Page 3 line 113 and page 6, lines 255 and 258: the terms “change-of-function” and “modification-of-function” are unclear. Do the authors mean “gain-of-function”?

Cancer-associated mutations of *SF3B1* are considered as neomorphic mutations that redirect splicing to novel splice sites or shift the ratio of alternative transcripts. Thus they do not formally correspond to “gain-of-function” mutations. To homogenize the manuscript, we replaced the term “Modification-of-function” (page 6 line 269) by change-of-function.

4. Page 3, line 122: we usually consider that NDD of genetic origin affect 3-4% of the population maximum. 10% probably includes very mild" forms and non-genetic forms of NDD.

We thank the reviewer for this remark. We modified the manuscript accordingly.

5. Page 3, line 130: please correct “RNU-2P” to “RNU2-2P”

We thank the reviewer for highlighting this mistake. We corrected the manuscript (RNU2-2).

Results

6. page 4, lines 148-154: The authors should briefly specify how the variants were identified and, if possible, provide some context on the nature and size of the cohorts used for variant discovery.

We thank the reviewer for this comment. Patients' DNA were sequenced primarily in a diagnostic setting or within various research cohorts at each center. We recruited the patients through the GeneMatcher platform. We modified the main text in the Methods section accordingly.

7. Could the authors indicate for each of the 26 patients if other variants have been identified by exome or genome analysis that could account for the patients' phenotype?

We thank the reviewer for this pertinent remark. This information is given in Table S1, in the line entitled "Other hits". We also modify the main text to add this precision (lines 158-160). Since the original submission, we were informed that P22 also carries a missense variant in RORA, that is inherited from his mother and considered as a variant of uncertain significance.

8. The clinical part lacks details about the nature of the variants associated with each feature. The authors may want to consider having the paragraph detailing the variants (lines 212-250)) before the clinical paragraphs so they can relate phenotypic features to specific variants. For instance, when discussing the three inherited cases, the exact variant details could be specified. This restructuring would also improve the clarity of genotype-phenotype correlations (Page 5, Lines 196–208).

We thank the reviewer for this comment. We modified the manuscript accordingly, which improves the clarity of the text. Based on the current sample size and the available clinical data, it is not yet possible to delineate a clear association between the molecular nature of the variant and the clinical features. Nevertheless, we proposed the hypothesis that missense variants could be associated with a more severe phenotype, compared to pLoF variants, as exemplified by more frequent congenital anomalies (i.e. microcephaly, heart malformation...). Concerning the inherited cases, we modified the text accordingly (lines 202-212).

9. It is unclear if dividing clinical features into multiple subheadings (head and neck, cardiac, etc.) is necessary. Removing the subheadings and linking these sections might improve readability and flow.

We modified the manuscript to remove the subheadings.

10. The abbreviation "pLoF" appears before it is defined (putative loss-of-function, Line 212). Consider defining it earlier.

We modified the main text accordingly.

11. How do the authors classify variants as putative loss-of-function (pLoF) in their genotype-phenotype correlations? Are they referring only to nonsense, canonical splice site variants (i.e., variants introducing premature termination codons) or do they also incorporate functional evidence? Additionally, please consider that missense variants can also result in loss of function (LoF), and LoF and gain of function (GoF) are not always mutually exclusive; a single variant may exhibit both effects.

We thank the reviewer for this comment. We chose to gather nonsense, canonical splice site and frameshift variants in the pLoF category. We are aware that missense variants can be responsible for loss, gain of function or both, but we showed that, for the variants described, they are not responsible for a loss of SF3B1 protein function. We added a sentence to precise the meaning of pLoF in the manuscript (lines 540-541).

12. Page 5, Line 218: The authors should be more specific when referring to pathogenicity criteria.

For example, rather than stating "pathogenicity predictors", they should explicitly list the predictive tools used (the reader only finds this out two sentences later).

We thank the reviewer for pointing out this issue. We added the pathogenicity predictors in the main text (line 168), which can also be found in Sup Table S2.

13. Page 5, lines 227-233: It is difficult to distinguish which variants are associated with cancer and which ones were identified in this study. Including an introductory paragraph earlier about somatic variants in SF3B1 found in cancer and their impact may improve clarity (see previous comment regarding the introduction).

We thank the reviewer for this comment. We modified the text accordingly, being more specific when citing cancer-associated variants.

14. Page 6, line 271: A similar issue with the skipping of exon 6 in DUSP11 and exon 11 in RBM5. Without prior context on these splicing events, the rationale is difficult to follow.

We thank the reviewer for this remark. We changed the text accordingly : "We then investigated specific exon skipping events known to be altered upon SF3B1 silencing, in particular in *DUSP11* (exon 6) and *RBM5* (exon 11)."

15. Page 6, line 272: Could the authors clarify whether they overexpress the SF3B1 p.Glu980* protein? If so, did they first verify, at least theoretically, that this variant escapes nonsense-mediated decay? Otherwise, the most likely consequence of this variant (and other pLoF variants) would be haploinsufficiency (without truncated protein produced).

We thank the reviewer for highlighting this point. We modified the text to clarify that we overexpressed the truncated protein. Due to the localization of the PTC, the p.Glu980* variant should produce transcripts that undergo NMD, leading to haploinsufficiency. This is why we considered this variant as a pLoF variant (cf paragraph on the proportion of variants). Nevertheless, we overexpressed the truncated p.Glu980 variant to evaluate whether, should the transcript escape NMD, the resulting truncated protein would retain any functional activity. Our results confirm the essential nature of the C-term part of the HEAT domain of SF3B1.

16. Similarly, have the authors investigated whether any of the missense variants identified in patients could impact SF3B1 splicing, such as by disrupting an ESE or creating/enhancing a novel splice site before overexpressing them?

We thank the reviewer for this pertinent comment. We used several splicing prediction tools (MaxEntScan, SpiP, dbSCSNV, SpliceAI) to test whether any of the missense variants could disrupt an ESE or enhance a novel splice site, and included the results in Sup table 2. Among the 16 missense variants analyzed, 10 were not predicted to have any effect by any of the computational tools employed. Three variants (E1029K, Y587N, and R397H) exhibited only a weak prediction of a potential ESE alteration and by SpiP only. The remaining three variants (E722K, M613V, and E906K) also yielded weak predictions, each supported by only one tool. We thus added the following sentence in the main text: "The use of several splicing prediction tools (MaxEntScan³⁴, SpiP³⁵, dbSCSNV³⁶, SpliceAI³⁷) does not provide strong evidence that missense variants may disrupt an ESE or favor usage of a novel splice site."

17. Page 6, lines 272-276: The same variant is referred to using different nomenclatures (p.Glu980*/p.Glu980Ter). Please homogenize.

We thank the reviewer for noticing this mistake. We homogenized using p.Glu980*.

18. Page 6, lines 278-279: The authors suddenly shift to the one-letter amino acid code (e.g., E980*). For clarity, the variant nomenclature should be consistent throughout the text.

We thank the reviewer for this comment. We decided to use the one-letter amino acid code throughout the text.

19. Page 12, line 566: Could the authors clarify what is meant by 'biological replicates' in the RNAseq experiments? Does this refer to independent cultures from the same immortalized cell clone (inducible cell lines) or were three different clones used? If the same clone was used, could variant-specific splicing be due to clonal effects? Please provide additional information in the methods section.

We thank the reviewer for this remark. In this study, biological replicates in the RNAseq experiments refer to independent cultures from the same inducible cell line pools. The cells were not cloned after transfection in order to avoid analysing clone-specific particularities. We specified it in the methods section: "Biological independent replicates (n=3), which refer to independent cultures from the same inducible cell line pool, were realised for each condition".

20. Could the authors provide further discussion on how variant-specific splicing might explain the phenotypic differences observed in patients?

We thank the reviewer for this suggestion. We added the following sentence in the discussion section, following the paragraph about somatic variants : "This suggests that each mutation may confer a unique splicing signature that influences disease phenotype. A similar scenario may occur for the NDD-associated SF3B1 variants, which might explain the phenotypic differences observed in patients."

21. While the rationale for using a cell model previously used to study splicing in cancer-specific variants is clear, could the authors discuss how the RNAseq results may be influenced by cell-type specificity?

We thank the reviewer for this pertinent comment. RNA splicing is highly regulated in a cell-type-specific manner, and splicing outcomes observed in one cellular context may not fully reflect those in other tissues or physiological states. Our choice of K562 model was guided by both its proven capacity to reveal splicing alterations linked to specific variants, as exemplified by cancer-associated SF3B1 variants, and the practical constraints associated with accessing disease-relevant tissues.

We agree that further validation in disease-relevant tissues or additional cell models would be valuable to comprehensively characterize the tissue-specific splicing effects of the variants, and we are currently exploring such approaches in ongoing work.

We added the following sentence in the results section “Considering RNA splicing is tightly regulated in a cell-type-specific manner, and that splicing outcomes observed in one cellular context may not fully recapitulate those in other tissues, further validation in additional cell models would be necessary to characterize the tissue-specific effects of the variants.” (lines 385-388)

Figures

22. Figure 1A: The heatmap is an effective way to visualize clinical features, but the meaning of the three clusters is unclear. It also seems that a significant portion of the clustering may be driven by missing data, rather than meaningful differences.

We thank the reviewer for this important comment. The first aim of the heatmap was to provide a descriptive overview of clinical similarities across patients, and the results led us to highlight an hypothetical difference between missense and pLoF variants. We are aware that missing data could influence the clustering, so we used the ward.D2 method to handle missing data. We also reanalysed the dataset after excluding features with >30% of missing values and we can see that the overall structure of the heatmap remained consistent. We modified the manuscript and Figure 1A accordingly, and included the second version of the heatmap in a supplementary Figure (S1)

Hierarchical clustering of clinical features (rows) of the cohort's patients (columns). Features with missing data in >30% of patients were excluded.

Reviewer #3 (Remarks to the Author):

In the manuscript by Uguen et al., the authors report several SF3B1 mutations found in a cohort of 26 patients with neurodevelopmental diseases (NDDs). Those mutations, mainly missense variants, are different from the hotspot mutations seen in cancer. Among the SF3B1 mutations seen in NDD patients, the authors focused on SF3B1 E722K, P780L, N829S to perform RNAseq analysis in K562 cells stably expressing these variants. The authors showed that SF3B1 E722K, P780L, N829S expression induces splicing changes compared to SF3B1 WT, to a lesser extent than the cancer associated K700E variant.

Major comments:

-The authors used K562 and HEK293T cells to study the splicing changes due to the described SF3B1 variants. These cancer cell lines are not an ideal model in this study due to the high mutational burden in these cancer cell lines and the non-physiologic SF3B1 expression. It would be ideal for the authors to use patient-derived fibroblasts, lymphoblastoid cell lines, or even primary human blood (or other cells) for the splicing analysis.

We thank the reviewer for this insightful comment. In this study, we chose to investigate the molecular effects of missense variants within a consistent genetic background, employing two distinct cell lines. This strategy was intended to minimize background-driven variability.

We agree that incorporating data from patient samples would be a major improvement. Despite the inherent challenges of obtaining patient-derived primary cells, we successfully collected peripheral blood samples from patients P1 (p.N829S) and P26 (p.P370L). Peripheral blood mononuclear cells (PBMC) were isolated and lymphocytes cultured for three days using the same standardized protocol as for controls (n = 14, NDD without any SF3B1 variant), prior to RNA extraction and sequencing. All samples were processed and cultured independently. Splicing analysis was subsequently performed using rMATS on both SF3B1 patient-derived and control samples. Notably, principal component analysis (PCA) revealed that the two patient samples clustered together, distinctly separate from the control group, suggesting a specific alternative splicing signature.

For the revised version, we included these results in Supplementary Figure 9, and modified the main text by adding the following paragraph: "To further support these findings, we examined alternative splicing profiles (using rMATS) in short-term cultured lymphocytes derived from two NDD patients (carrying variants p.N829S and p.P370L) in comparison to 14 controls. Remarkably, principal component analysis (PCA) showed that the two patient samples clustered together and separately from the controls, which strongly suggests a distinct splicing signature associated with p.N829S and p.P370L variants (Figure 4F and Sup Figure 9)".

-The authors claim that SF3B1 variants described in this manuscript are associated with NDDs, but they do not include in their analysis a comparison with healthy controls, or NDD patients with SF3B1 WT. What is the frequency of the reported SF3B1 mutations in healthy controls, or NDD patients from HUGODIMS consortium?

We thank the reviewer for this remark. This information can be found in line 163 where we indicate that all variants were absent from general population databases, namely gnomAd. Regarding the frequency in patients from the HUGODIMS consortium; except for patient 1, no other variant in SF3B1 was identified.

-The three variants that the authors focused on for the RNAseq analysis are single, De Novo variants. However, in the cohort described in this study there was three inherited SF3B1 variants, namely p.(Glu809AspfsTer7), p.(Val695TrpfsTer34) and p.(Arg166Ter) and two recurrent variants, namely P370S/L and R939C. Analysis of the splicing changes due to these inherited or recurrent SF3B1 variants could have been a stronger argument that SF3B1 is involved in NDDs.

We thank the reviewer for these constructive comments. As the majority of patients (two-thirds) carry missense variants, we chose to focus specifically on this subset within the scope of the present study.

Given the high number of missense variants identified, we had to select the variants included in the RNA sequencing analyses conducted on K562 cells. Still, following RNA sequencing, we were able to extend the identification of missplicing events (CRNDE, PQBP1, ENOX2) to more missense variants, including T703P and L536I.

The recurrent variant **p.R939C** was incorporated within the splicing-targeted study; however, it was not selected for the RNA-seq analysis, as its recurrent status had not been established at the time the experiments were conducted. The next phase of this research will certainly include a comprehensive analysis of this recurrent variant.

Regarding the second recurrent variant position : for the revised version of the manuscript, we performed splicing assays for both **P370S and P370L**, and showed that these two variants do not inactivate SF3B1 function (please see Sup Figure 4). Moreover, we show that none of the substitutions (P370S and P370L) leads to missplicing typical of K700E (new figure 3E). Finally, peripheral blood samples were successfully obtained from patient P26 (p.P370L). Remarkably, as described above, RNA sequencing of cultured lymphocytes derived from this patient substantiated that this SF3B1 variant does induce alterations in the alternative splicing profile relative to controls without SF3B1 mutation.

For the revised version, we added the splicing data for P370S/L (DUSP11 and RBM5 splicing assay) in Supplementary Figure 4 and in Figure 3E for the cancer-associated splicing events. We also incorporated the PCA performed on PSI values of significant alternative splicing events from short-term cultured lymphocytes (Figure 4G and sup Figure 9)

-The authors only mention in line #444 that patient 1 also harbors LARP1 mutation. A systematic description of all the genetic mutations, or absence of it, in the 26-patient cohort is important to associate SF3B1 with NDDs in this cohort.

We thank the reviewer for this remark. This information is given in Table S1, in the line entitled "Other hits". We modified the main text to add this information (first section of Results).

Reviewer #4 (Remarks to the Author):

Other modifications :

In addition to the modifications suggested by the reviewers, we added the validation of HSPBP1 splicing event (A3'SS) and corrected a few typos.

We also added more information about the P22 individual in the subheading entitled "*inheritance*".

Please find here the list of modifications made to the figures:

- Figure 1A: modification of the map
- Figure 2A: correction of a typo
- Figure 2E: replacement of the E722(K) zoom-in by the R939(C) zoom-in.
- Figure 3E: incorporation of the results for P370S and P370L
- Figure 4: inclusion of panel G (PCA on splicing events from patients lymphocytes)
- Figure 5B and 5D: inclusion of the validation of HSPBP1 splicing event (A3'SS)
- Addition of a Supplementary Figure (S1) entitled "Hierarchical clustering of clinical features (rows) of the cohort's patients (columns). Features with missing data in >30% of patients were excluded."
- Extension of the Supplementary Figure S4 (previously S3), with data on the molecular impact of *SF3B1* missense variants P370S and P370L.
- Addition of a Supplementary Figure (S9) entitled "Principal component analysis performed on PSI values of significant alternative splicing events from short-term cultured lymphocytes."